# The translation elongation factor eEF1A1 couples transcription to translation during heat shock response

Maria Vera[1,2,4,5], Bibhusita Pani[1], Lowri A Griffiths[3], Christian Muchardt[2], Catherine M Abbott[3], Robert H Singer[4,5], Evgeny Nudler[1,6]*

[1]Department of Biochemistry and Molecular Pharmacology, New York University School of Medicine, New York, United States; [2]Département de Biologie du Développement et Cellules Souches, Institut Pasteur, CNRS URA2578, Paris, France; [3]Medical Genetics Section, Molecular Medicine Centre, Institute of Genetics and Molecular Medicine, Western General Hospital, University of Edinburgh, Edinburgh, United Kingdom; [4]Department of Anatomy and Structural Biology, Albert Einstein College of Medicine, New York, United States; [5]Gruss-Lipper Biophotonics Center, Albert Einstein College of Medicine, New York, United States; [6]Howard Hughes Medical Institute, New York University School of Medicine, New York, United States

**Abstract** Translation elongation factor eEF1A has a well-defined role in protein synthesis. In this study, we demonstrate a new role for eEF1A: it participates in the entire process of the heat shock response (HSR) in mammalian cells from transcription through translation. Upon stress, isoform 1 of eEF1A rapidly activates transcription of HSP70 by recruiting the master regulator HSF1 to its promoter. eEF1A1 then associates with elongating RNA polymerase II and the 3'UTR of HSP70 mRNA, stabilizing it and facilitating its transport from the nucleus to active ribosomes. eEF1A1-depleted cells exhibit severely impaired HSR and compromised thermotolerance. In contrast, tissue-specific isoform 2 of eEF1A does not support HSR. By adjusting transcriptional yield to translational needs, eEF1A1 renders HSR rapid, robust, and highly selective; thus, representing an attractive therapeutic target for numerous conditions associated with disrupted protein homeostasis, ranging from neurodegeneration to cancer.

*For correspondence: evgeny.nudler@nyumc.org

## Introduction

All organisms respond to environmental stress by activating genes that encode molecular chaperones and other cytoprotective heat shock proteins (HSPs) in a process known as the heat shock response (HSR) (*Lindquist and Craig, 1988*). In mammalian cells, HSR is regulated at the level of transcription by the heat shock factor 1 (HSF1) (*Guertin and Lis, 2010*; *Anckar and Sistonen, 2011*). In coordination with their augmented transcription, the stability, transport, and translation of HSP mRNAs (especially HSP70) are sharply increased during HSR, accounting for the robust production of HSPs (*Theodorakis and Morimoto, 1987*; *Zhao et al., 2002*). HSP70 is the best-studied example, which is among the most inducible HSPs in terms of promptness and robustness of the response (*Theodorakis and Morimoto, 1987*; *Lindquist and Craig, 1988*; *Cuesta et al., 2000*). The mechanism of such a rapid and synchronized reaction to stress remains unknown, but is particularly remarkable considering that upon stress, the majority of other genes are inhibited at the level of transcription, RNA export, and translation (*Laroia et al., 1999*; *Gallouzi et al., 2000*; *Mariner et al., 2008*).

Our previous in vitro studies demonstrated that translation elongation factor eEF1A enhances the binding of HSF1 to a DNA containing an artificial HSE sequence via a mechanism that requires a non-coding RNA (*HSR1*) (*Shamovsky et al., 2006*). eEF1A1 has a well-defined role in protein synthesis, it

**eLife digest** Living cells must be able to withstand changes in the environment. For example, if there is a sudden increase in temperature—which could damage proteins or other molecules—most cells can respond with the 'heat shock response'. In humans and other mammals, a single protein called heat shock factor 1 triggers the production of numerous heat shock proteins that protect the cell from the detrimental effects of high temperatures. Most heat shock proteins protect cells by binding to, and stabilizing, other molecules in the cell; this prevents these molecules from being damaged or from aggregating and allows them to continue to function as normal.

A protein called eEF1A1 is involved in the final stages of protein production and also enhances the function of heat shock factor 1 during the heat shock response. To make a protein, an enzyme called RNA polymerase transcribes DNA in the nucleus of the cell into a messenger RNA molecule that then exits the nucleus and binds to a ribosome. This molecular machine then translates the messenger RNA sequence into a protein by joining together individual building blocks called amino acids in the correct order. Like other elongation factors, eEF1A1 helps to select the amino acids that match the sequence of the messenger RNA template. However, it was unclear how eEF1A1 helped to protect cells during the heat shock response.

Nudler et al. have now engineered cells—from humans and mice—that make less of the eEF1A1 protein than normal. These cells had enough of this protein to support their growth and development under normal conditions, but not enough to help during the heat shock response. When these cells are subjected to a sudden increase in temperature, they fail to produce a sufficient amount of major heat shock proteins. Heat shock factor 1 is needed to transcribe the genes that encode these heat shock proteins, and Nudler et al. found that eEF1A1 must bind to heat shock factor 1 and then to a moving RNA polymerase for these genes to be transcribed efficiently. Moreover, the eEF1A1 protein was shown to bind to and stabilize the heat shock proteins' messenger RNAs, and aid their export from the nucleus and their binding to the ribosome.

These newly discovered roles for eEF1A1 during the heat shock response highlight this elongation factor as a promising drug target for treating diseases where protein folding goes awry, for example in Alzheimer's or Parkinson's disease. In adults, neurons do not make enough eEF1A1, and Nudler et al. suggest that enabling these cells to make more of this protein could help to treat a range of neurodegenerative conditions.

delivers aminoacylated tRNAs to the A site of ribosome during translation elongation in a GTP-dependent manner (*Mateyak and Kinzy, 2010*; *Li et al., 2013*). While the majority of eEF1A1 associates with the cytoplasmic cytoskeleton and translational machinery (*Ejiri, 2002*), eEF1A1 is able to enter the nucleus to mediate the export of tRNAs and SNAG-containing proteins under normal growth conditions (*Bohnsack et al., 2002*; *Calado et al., 2002*; *Mingot et al., 2013*). Also, eEF1A can directly bind mammalian mRNAs (*Liu et al., 2002*; *Mickleburgh et al., 2006*) and mediate stability of viral and cellular RNAs by binding to their 3' regions (*Yan et al., 2008*; *Li et al., 2013*).

eEF1A expression in mammalian cells happens in one of two isoforms, eEF1A1 and eEF1A2. These two isoforms are 98% similar at the amino acid level and share the same canonical function. Each isoform is the product of expression of a specific locus and shows a distinct expression pattern except in transformed cells, which express both. While the majority of cells express eEF1A1 isoform, adult neuronal and muscle cells express eEF1A2 (*Newbery et al., 2007*; *Abbott et al., 2009*). In this study, we report that eEF1A1 isoform, but not the tissue-specific variant eEF1A2 is required for the HSP70 induction during heat shock. It mediates transcription-activation of HSP70 mRNA, its stability, transport, and translation, thereby synchronizing HSP70 transcriptional output to translational needs.

## Results

### Transcription of HSPs genes is impaired in cells knocked down for eEF1A1

To determine the role of eEF1A1 in HSR in vivo, we designed a strategy for its selective and efficient knockdown without compromising overall translation and cell viability. eEF1A is one of the most

abundant cellular proteins, compromising 1–3% of total cytosolic proteins, and is present in large excess to its protein synthesis partners (molar ratios for eEF1A:eEF1B and eEF1A:ribosomes of 10:1 and 25:1 respectively) (*Slobin, 1980*). Therefore, a ~70% knockdown of eEF1A1 did not compromise cell viability under non-stress conditions and overall translation under control or mild-heat shock conditions (*Figure 1—figure supplement 1A,B*). The levels of eEF1A1 and HSF1 (positive control), but not those of GAPDH, were greatly reduced after efficient transfection of human breast cancer cells (MDA-MB231), primary human fibroblasts (WI38), immortalized mouse embryonic fibroblast (MEFs), or mouse NSC34 motor spinal cord/neuroblastoma fusion cells, with specific and un-related sets of siR-NAs (*Figure 1A* [siRNA pair A], *Figure 1—figure supplement 2A* [siRNA pair B], *Figure 1—figure supplement 3A,B* [siRNA C]). After heat shock, knock down of eEF1A1 (~30% of control remaining) or HSF1, (<25% of control remaining), markedly reduced the induction of HSP70 (*Figure 1A*, *Figure 1—figure supplement 2A*, *Figure 1—figure supplement 3A,B*) and HSP27 (*Figure1—figure supplement 3C*), and resulted in loss of thermo-tolerance (*Figure 1B*). To rule out the possibility that the loss of induction of HSPs in eEF1A1-deficient cells was potentially due to overall compromised translation, de novo protein synthesis was analyzed in a pulse-chase experiment with [35S]-methionine. We observed a general decrease in protein synthesis after HS, but no significant difference between mock-transfected cells and cells depleted of either HSF1 or eEF1A1 (*Figure 1—figure supplement 1A*). Consistently, HSP70 and HSP27 mRNA levels were significantly decreased in eEF1A1-knocked down heat-shocked cells (*Figure 1C*, *Figure 1—figure supplement 2B*, *Figure 1—figure supplement 3D,E*). Impaired mRNA induction upon heat shock due to eEF1A1 deficiency was observed for several classes of HSP genes (*Figure1—figure supplement 3F*). It is known that HSP70 mRNA induction occurs upon chemical inhibition of translation. To verify that the action of eEF1A1 is independent of its role in translation inhibition, we tested eEF1A1 knockdown conditions in combination with inhibitors of translation elongation. eEF1A1-depleted cells treated with the HSR-stimulating inhibitors of translation, cycloheximide (CHX) or doxycycline (DOX) failed to accumulate HSP70 mRNA (*Figure 1—figure supplement 3G*). In contrast, knocking down eEF1A2 isoform had no effect on HSP70 mRNA induction or protein amount (*Figure 1—figure supplement 4A,B*). These results demonstrate that isoform 1 of eEF1A specifically controls HSPs gene expression irrespective of its role in translation.

Analyses of the distribution of RNA polymerase II (RNAPII) along the HSP70 gene as a function of the presence of eEF1A1 supported this conclusion, arguing that eEF1A1 is an activator of transcription upon stress. Chromatin immunoprecipitation (ChIP) and run-on assays demonstrated that knock down of eEF1A1 reduced recruitment of RNAPII throughout the HSP70 gene, indicative of both impaired RNAPII initiation and elongation during heat shock (*Figure 1D*, *Figure 1—figure supplement 2C*, *Figure 1—figure supplement 5A–C*). Under the same conditions, there was no effect of eEF1A1 depletion on RNAPII at the GAPDH gene (*Figure 1D*, *Figure 1—figure supplement 2C*). Knock down of eEF1A1 did not change the relative occupancy of RNAPII on HSP70 and GAPDH genes under non-heat shock conditions (*Figure 1D*).

We used single molecule fluorescence in situ hybridization (smFISH) and airlocalize software (*Lionnet et al., 2011*) to assess the kinetics of HSP70 transcription (from control growth conditions to 2 hr of HS) and quantify the number of HSP70 transcription sites (TS) per cell and the transcription activity per TS in the presence or absence of eEF1A1. eEF1A1 knock down was monitored in immortalized mouse embryonic fibroblast (MEFs) from the homozygous *Actb*-MBS mouse strain (*Lionnet et al., 2011*) expressing eEF1A1 fused to cherry and Flag, by the decrease in cherry detection (*Figure 1E*, *Figure 1—figure supplement 1D*, *Figure 1—figure supplement 6A*). At 30 min of HS, 80% of cells showed a strong nuclear FISH signal that accounted for active HSP70 transcription in at least 1 of its 4 alleles (MEFs are tetraploid) (*Figure 1E*, *Figure 1—figure supplement 1D*, *Figure 1—figure supplement 6B,C*). Cells knocked down for eEF1A1 showed half (pair A) or third (pair B) of the intensity of transcription per cell (*Figure 1F*, *Figure 1—figure supplement 1E*). This result is in perfect correlation with the quantification of HSP70 mRNA, nuclear run-on and ChIP experiments. This decrease of transcription intensity per cell is due to both a decrease in the intensity quantified per TS (*Figure 1F*, *Figure1—figure supplement 1D*) and a reduction in the number of TS per cell from a mean of 3 TS per cell in non-transfected cells to 2.6 TS in cells knocked down for eEF1A1 (siRNA pair A) and from 2.8 to 2.1 (siRNA pair B).

## eEF1A1 enhances the binding of HSF1 to HSP70 promoter

To gain further insight in the mechanism by which eEF1A1 regulates transcription of HSP genes, we explored the activation of HSF1. Upon stress, HSF1 quickly trimerizes, acquires the ability to bind HSE

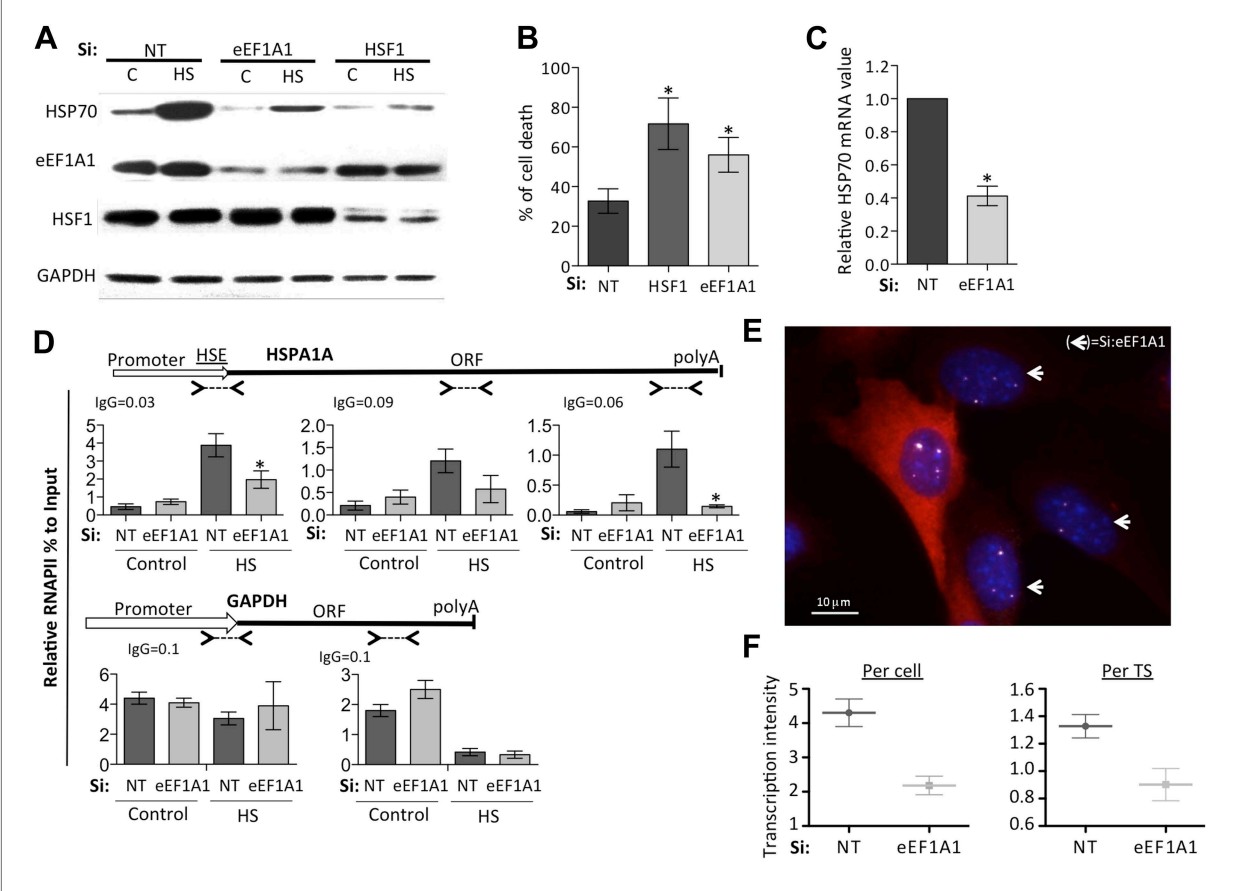

**Figure 1**. eEF1A1 regulates HSP70 expression and thermotolerance. (**A**) Knock down of eEF1A1 decreases HSP70 protein expression upon heat shock. Western blots of HSP70, eEF1A1, HSF1, and GAPDH from total cell lysates of MDA-MB231 cells transfected with siRNA (pair A) against eEF1A1 or HSF1, or mock-transfected (Si:NT with no target). Control (C)—unstressed cells kept at 37°C. Heat shock (HS)—cells kept for 1 hr at 43°C followed by 6 hr of recovery at 37°C. (**B**) HSF1- or eEF1A1-knocked down cells are less thermo-tolerant. Cell death was quantified by FACS analysis after propidium iodide staining. Thermotolerance was induced in mock-transfected (Si:NT) or eEF1A1/HSF1-depleted cells by two heat treatments (1 hr at 43°C, 12 hr at 37°C, and 1 hr at 45°C), followed by 12 hr of recovery. Data from three independent experiments are presented as the mean ± SEM. *p < 0.05. (**C**) Knock down of eEF1A1 decreases HSP70 mRNA expression during heat shock. Total RNA from heat shocked MDA-MB231 cells was reverse-transcribed with random hexamer primers followed by quantification of GAPDH and HSP70 mRNAs by QPCR. HSP70 mRNA values were normalized to that of GAPDH. Bars represent the amount of HSP70 mRNA in cells depleted of eEF1A1 relative to that obtained for mock-transfected cells (Si:NT). 1 is the value of HSP70 mRNA in Si:NT cells at 1 hr of heat shock. Data from three independent experiments are presented as the mean ± SEM. *p < 0.05. (**D**) eEF1A1 controls RNAPII occupancy at the HSP70 gene after stress as determined by ChIP-QPCR. Schematic of the HSPA1A locus is shown at the top. Arrowheads indicate the regions amplified by QPCR. Panels show the effect of eEF1A1 knock down (si:eEF1A1) on RNAPII occupancy relative to input Ct value under non-heat shock conditions (control) and after 30 min of heat shock. The relative value for the IgG is indicated for each of the PCR fragments on top of the plot. Values below that of the IgG mean non-specific binding. Mock-transfected cells (Si:NT). By comparison, GAPDH showed no change with si;eEF1A1. Data from three independent experiments are presented as the mean ± SEM. (*p < 0.05). (**E**) eEF1A1 mediates HSP70 transcription upon HS. MEFs were infected with a lentivirus expressing Cherry-eEF1A1. eEF1A1 expression was knocked down by siRNA. At 30 min after HS, cells were fixed and HSP70 mRNA detected by FISH. Nuclei were stained with DAPI. Merged images show Cherry-eEF1A1 in red, HSP70 TS in gray and nucleus in blue. White arrows indicate cells knocked down of eEF1A1. Bar = 10 microns. (**F**) eEF1A1 mediates HSP70 transcription upon HS. Plots are the quantification of the intensity of HSP70 transcription, per cell or per TS, detected by FISH and quantified by airlocalize. A total of seventy cells per condition were analyzed from three independent experiments. NT = non-transfected cells. eEF1A1 = cells transfected with siRNA for eEF1A1.

The following figure supplements are available for figure 1:

**Figure supplement 1**. Knock down of eEF1A1 does not affect general translation or cell viability.

**Figure supplement 2**. eEF1A1 regulates HSP70 expression (all experiments in this figure were done with an alternative pair of siRNAs [pair B] to rule out off-target effects).

*Figure 1. Continued on next page*

*Figure 1. Continued*

**Figure supplement 3**. Knock down of eEF1A1 reduces HSPs expression in mouse and human cells.

**Figure supplement 4**. eEF1A2 does not support HSR.

**Figure supplement 5**. Knock down of eEF1A1 decreases overall transcription of HSP70.

**Figure supplement 6**. Characterization of MEFS expressing Cherry-Flag-eEF1A1.

(*Baler et al., 1993*; *Sarge et al., 1993*; *Guertin and Lis, 2010*; *Anckar and Sistonen, 2011*), and reactivates paused RNAPII via a mechanism involving recruitment of the Mediator complex, the CTD kinase P-TEFb (*Park et al., 2001*; *Ni et al., 2004*; *Shen et al., 2009*) and the SWI/SNF chromatin remodeling complex (*Corey et al., 2003*). In MDA-MB231 cells, as in other cell lines, HSF1 is localized in the nucleus under normal growth conditions. In mammalian cells, HSE binding activity assessed by EMSA is entirely due to HSF1 (*Figure 2D*, *Figure 2—figure supplement 1B*). This binding activity was reduced in the protein extract (10 μg) from eEF1A1-knocked down cells (*Figure 2A*). Likewise, knock down of eEF1A1 also resulted in a significant reduction in recruitment of HSF1 to the HSP70 promoter, as determined by ChIP assays (*Figure 2B*). The low levels of HSF1 binding to HSP70 promoter under control conditions did not change in cells knocked down of eEF1A1. These results indicate that eEF1A1 enhances the binding of HSF1 to HSP70 promoter and suggest an interaction between eEF1A1 and HSF1 during HS.

Consistent with the above observations, HSF1 co-immunoprecipitated with eEF1A1 upon HS (*Figure 2C*, *Figure 2—figure supplement 1A*). Moreover, eEF1A1 formed a ternary complex with HSF1 and DNA. Heat shock or arsenic treatments induce HSF1 DNA binding, as detected by EMSA. Under these conditions, anti-HSF1 and anti-eEF1A1, but not mouse IgG, super-shifted the HSF1–DNA complex (*Figure 2D*, *Figure 2—figure supplement 1B*). We noted that the super-shifts with anti-eEF1A1 and anti-HSF2 antibodies were similar. Considering that HSF2 binds HSE only when forming a hetero-trimer with HSF1 (*Ostling et al., 2007*), this observation indicates further that eEF1A1 associates directly with DNA-bound HSF1. Moreover, in vitro binding of recombinant eEF1A1 to a DNA fragment of HSP70 promoter suggests direct and specific, albeit a relatively weak, interaction of eEF1A1 to HSP70 promoter (*Figure 2E*). We further confirmed the presence of eEF1A1 at the HSP70 locus by ChIP-QPCR experiments with an eEF1A1-specific antibody (*Figure 2F*, *Figure 2—figure supplement 1C*). Occupancy of HSP70 and HSP27 promoters by eEF1A1 was significantly higher (p < 0.05) than at the GAPDH promoter. These results demonstrate that eEF1A1 binds specifically to promoters of at least two HSP genes before and after stress.

Together, the above results demonstrate that eEF1A1 recruits HSF1 to the HSP70 promoter upon stress and triggers HSR by functioning as a transcriptional co-activator.

## eEF1A1 localizes to HSP70 transcription sites and interacts with RNAPII

Observations described above strongly suggest that eEF1A1 is in the nucleus of the cell at the TS of HSP70. To gain further insight in the cellular localization of eEF1A1 regarding HSP70 mRNA, we used fluorescence in situ hybridization (FISH). We used the MEFs expressing eEF1A1 fused to Cherry and Flag (*Figure 1—figure supplement 5A*), and we infected human diploid fibroblasts (TIGs) to also express recombinant eEF1A1 protein fused to Cherry and Flag (*Figure 3—figure supplement 1*). MEFs were subjected to different times of HS (from 0 to 60 min) before visualizing HSP70 mRNA. From 30 to 60 min of HS ~80% of the cells showed a strong nuclear FISH signal that accounted for active HSP70 transcription (*Figure 3A*, *Figure 1—figure supplement 5B,C*). After 30 min of heat shock, we quantified the dots representing Cherry-eEF1A1 concentrated at several HSP70 TS (*Figure 3A*, *Figure 3—figure supplement 2*). Percentages of HSP70 TS with a positive signal for Cherry-eEF1A1 were quantified by airlocalize™ software (*Lionnet et al., 2011*). The percent of Cherry-eEF1A colocalizing with the HSP TS increased from 27% to −64% for 30 to 60 min (*Figure 3C*). Using probes for β-actin TS instead of those for HSP70 (*Figure 3B*), we did not see any co-localization with eEF1A nuclear dots. The Cherry-eEF1A dots were not a result of bleed through of the FISH signal because when HSP70 FISH was performed in MEFs that do not express Cherry-eEF1A1, nuclear dots in the red channel were

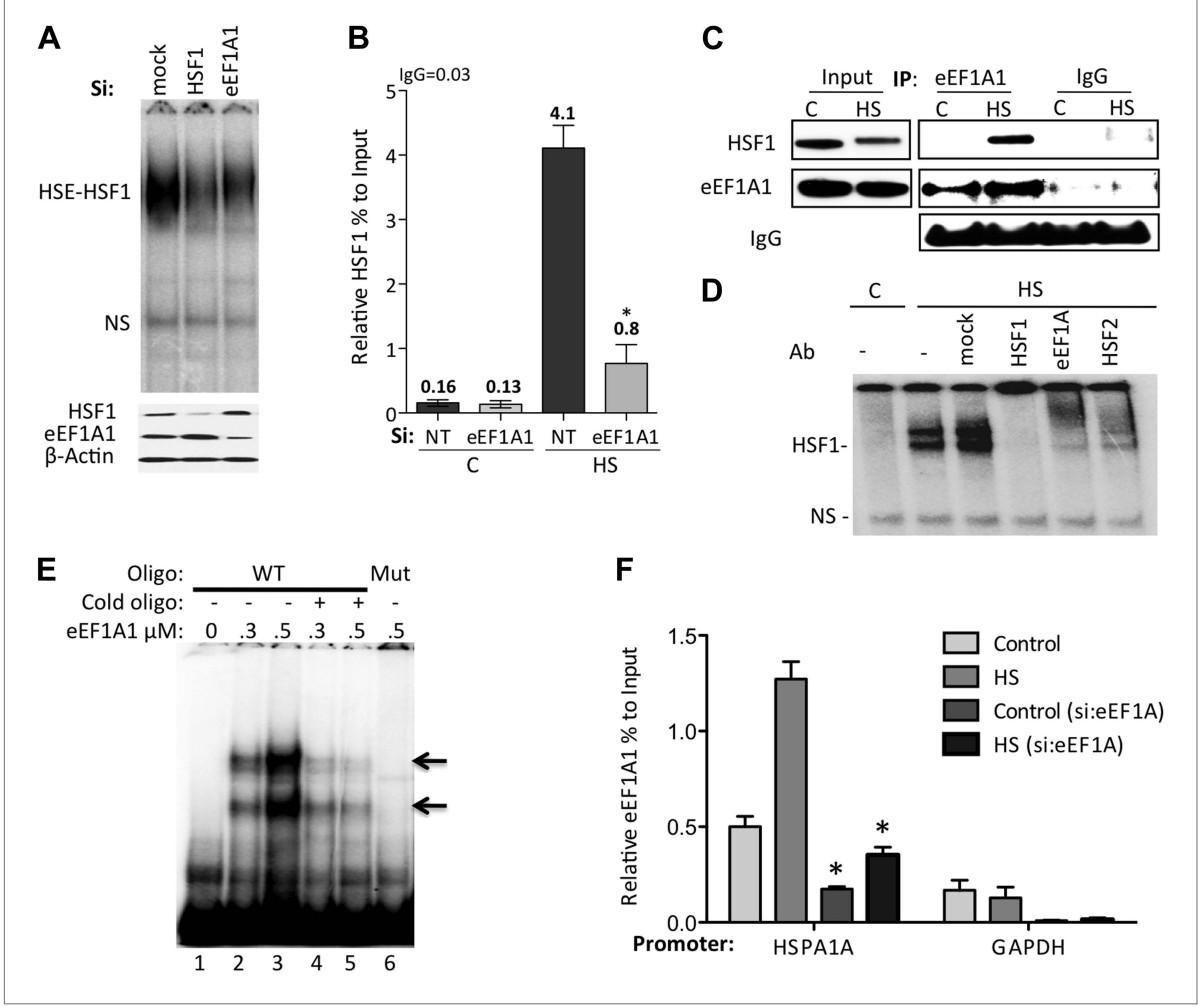

**Figure 2**. eEF1A1 mediates HSF1 recruitment to HSP70 promoter. (**A**) eEF1A1 enhances HSF1 DNA binding upon heat shock. Mock-transfected, eEF1A1 or HSF1-knocked down MDA-MB231 cells were heat-shocked for 30 min at 43°C and analyzed by HSF1-HSE EMSA (top panel). 10 μg of total protein were loaded in the EMSA. HSF1, eEF1A1, and GAPDH levels were determined by immunoblotting (lower panel). (**B**) eEF1A1 is required for HSF1 promoter binding in vivo. ChIP-QPCR was performed on mock-transfected (Si:NT) or eEF1A1-knocked down (Si:eEF1A1) cells. Panel shows the effect of eEF1A1 depletion on HSF1 occupancy at the HSP70 promoter (relative to the input Ct value) under non-heat shock conditions (C) and after 30 min of heat shock (HS). The reference value for IgG control is indicated on top of the plot. Values below those numbers mean non-specific binding. Data from three independent experiments are represented as the mean ± SEM. (*p < 0.05). (**C**) Stress-induced formation of the eEF1A1-HSF1 complex in vivo. Extracts from unstressed (C) or heat-shocked (HS) MDA-MB231 cells IPed with eEF1A1 antibody or IgG. IP samples or total protein (Input) were subjected to SDS-PAGE and immunoblotting. (**D**) eEF1A1-HSF1 complex formation at HSE. Panels show the super-shift of HSF1-HSE EMSA caused by specified antibodies. MDA-MB231 cells were heat-shocked for 20 min at 43°C (HS). Extracts were incubated with antibodies to HSF1, HSF2 (positive control), or eEF1A1 antibodies, or IgG (mock). HSE-HSF1 indicates specific binding of HSF1 to labeled HSE; NS—a non-specific band. (**E**) Direct binding of eEF1A1 to HSP70 promoter DNA. Radiolabeled fragment of the region −141 to −91 of the human HSP70 promoter was incubated with purified eEF1A1 (lanes 1–3), chased with fivefold molar excess of cold oligonucleotide (lanes 4 and 5), or mutant fragment of the same region (lane 6). Arrows mark the specific eEF1A1 shift. (**F**) eEF1A1 binds the HSP70 promoter before and after stress. Mock transfected or eEF1A1-knocked down (Si:eEF1A) MDA-MB231 cells were kept at 37°C or heat-shocked (HS) for 20 min at 43°C. Chromatin IP with eEF1A1 or IgG antibodies and amplified by PCR for the HSP70 (HSPA1A) and GAPDH promoters. Relative value of the control IgG vs input for HSPA1A is 0.04 and for GAPDH is 0.05. Data from three independent experiments are presented as the mean ± SEM. *p < 0.05.

The following figure supplement is available for figure 2:

**Figure supplement 1**. eEF1A1 interacts with HSF1 at HSE upon stress and binds HSP27 promoter.

rarely apparent, and when FISH was not performed the Cherry-eEF1A1 dots remained evident (***Figure 3—figure supplement 3***) The presence of eEF1A1 at HSP70 TS was further confirmed in a human fibroblast cell line (TIG) (***Figure 3—figure supplement 1***).

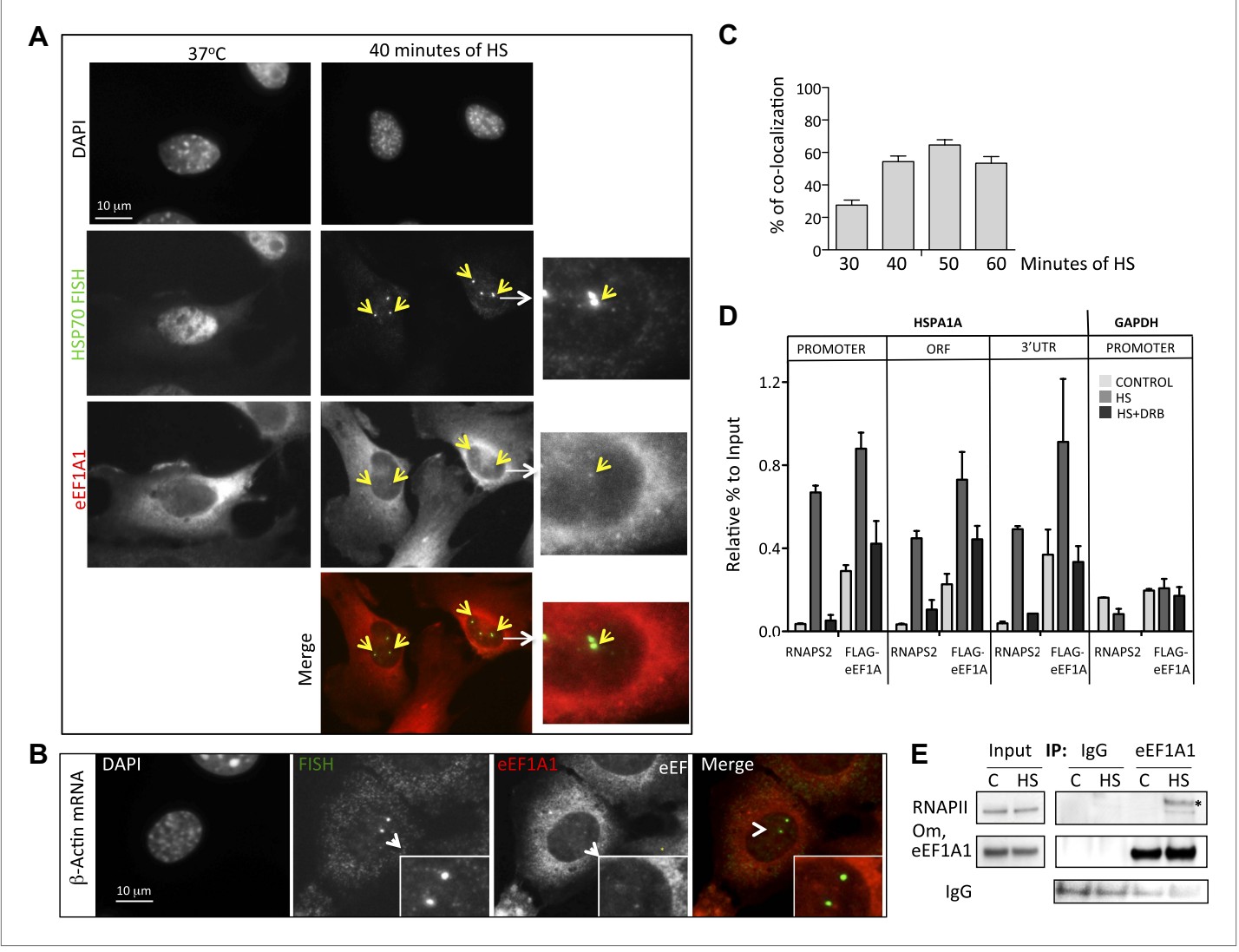

**Figure 3**. eEF1A1 localizes at HSP70 TS and interacts with RNAPII upon HS. (**A**) eEF1A1 localizes to HSP70 TS upon HS. MEFs were infected with a lentivirus expressing Cherry-eEF1A1. At the indicated times after HS, cells were fixed and HSP70 mRNA detected by FISH. Nucleus stained with DAPI. Merged images show HSP70 FISH in green and cherry-eEF1A1 in red. The nascent mRNA signal is much brighter than the Cherry-eEF1A1 because there were many nascent chains each detected with 48 probes (48 fluors), compared to a single fluorescent protein for eEF1A1, further diminished by fixation. Yellow arrows indicate TS for HSP70 where Cherry-eEF1A1 was also detected. Inset location indicated by white arrowheads. n = total number of cells analyzed from three independent experiments. (%) = percentage of TS with co-localization for eEF1A1. Bar = 10 microns. (**B**) eEF1A1 localizes in nuclear dots. MEFs were infected with a lentivirus expressing Cherry-eEF1A1. At 1 hr of HS cells were fixed and FISH was carried out to detect β-actin mRNA. Nucleus DAPI stained. Merged images show HSP70 FISH in green and Cherry-eEF1A1 in red. Arrowhead = inset location. Note that nuclear localization of Cherry-eEF1A1 does not coincide with the β-actin mRNA TS. (**C**) Quantification of co-localization between eEF1A1 and HSP70 TS. Percentages of co-localization were quantified by airlocalize software at the indicated HS times. Average of three different experiments. Total n = (80–90) cells per time point. (**D**) DRB decreases RNAPIIS2 and eEF1A1 occupancy within the HSP70 gene in HS cells. Data are the mean ± SEM from three independent experiments. MEF cells expressing eEF1A1 tagged with Cherry and Flag were kept under normal growth conditions (control) or heat-shocked for 40 min at 43°C (HS) or treated with 100 μM DRB for 15 min followed by HS (HS+DRB). ChIP was performed using antibodies for RNAPII phosphorylated at Ser2 (RNAPS2) and Flag eEF1A1 followed by QPCR with the indicated primers. (**E**) eEF1A1 binds RNAPII during heat shock. Extracts from unstressed (C) or heat-shocked (HS) MDA-MB231 cells were IP with an eEF1A1 antibody or IgG (mock). IP samples or total protein (Input) were subjected to SDS-PAGE and immunoblotting with RNAPII and eEF1A1 antibodies. (*) Indicates the hyperphosphorylated form of RNAPII.

The following figure supplements are available for figure 3:

**Figure supplement 1**. eEF1A1 localizes to HSP70 TS in the human fibroblast cell line TIG.

*Figure 3. Continued on next page*

*Figure 3. Continued*

**Figure supplement 2**. eEF1A1 localizes to HSP70 TS upon HS.

**Figure supplement 3**. eEF1A1 localizes to HSP70 TS upon HS-negative controls.

**Figure supplement 4**. DRB decreases RNAPII and eEF1A1 occupancy within the HSP70 gene upon HS.

Since several molecules of eEF1A need to accumulate in a discrete site to be detected by microscopy, we hypothesized that the presence of eEF1A at HSP70 TS not only depends on its interaction with HSF1, but also on the increase in transcription. ChIP analysis of heated MEF cells (expressing Flag-eEF1A1 or not) showed that eEF1A1 occupancy of HSP70 gene increases in the ORF and 3′UTR upon HS, as does RNAPIIS2 and RNAPII (*Figure 3D*, *Figure 3—figure supplement 4A,B*). When RNAPII elongation was inhibited by DRB treatment before HS, RNAPIIS2 occupancy on HSP70 gene was abolished (*Figure 3D*). Interestingly, the eEF1A1 occupancy inside HSP70 locus was reduced (p = 0.0585) as it was for total RNAPII (p = 0.0316) (*Figure 3D*, *Figure 3—figure supplement 4A,B*).

This result together with the fact that eEF1A1 co-immunoprecipitates with the RNAPII (*Figure 3E*) suggests that eEF1A1 interacts with the transcription elongation complex during HSP70 transcription.

## eEF1A1 binds to the 3′UTR of HSP70 and regulates its stability

Detection of eEF1A1 at the 3′ end of HSP70 gene suggests a role for this protein beyond transcription. Thus, we investigated the interaction between eEF1A1 and HSP70 mRNA in the soluble chromatin fraction of control and HS cells. HSP70 mRNA association with eEF1A was detected by RNA-IP in HS cells (*Figure 4A*). In RNA-IP experiments from total cell lysates, we also detected HSP70 mRNA associated with eEF1A1 and other known HSP70 mRNA-binders, poly(A)-binding protein 1 (PABP1) and nuclear pore complex (NPC) protein TPR1 (*Skaggs et al., 2007*; *Figure 4B*).

We then used RNA-EMSA to test for a direct interaction between eEF1A1 and HSP70 mRNA. eEF1A1 altered the mobility of the 3′UTR HSP70 mRNA probe, but not that of the 5′ portion of the HSP70 mRNA or of an unrelated RNA (200 nt β-actin ORF) (*Figure 4C*). These observations prompted us to investigate a role for eEF1A1 in stabilizing HSP mRNA. We monitored HSP70 and GAPDH mRNA decay after inhibition of transcription by actinomycin D (*Figure 4D*, *Figure 4—figure supplement 1A*). After 30 min of HS treatment, the rate of HSP70 mRNA decay increased substantially in eEF1A1-depleted cells, as compared to mock-transfected cells. In contrast, no change in the rate of GAPDH mRNA decay was observed.

To roughly map the region within HSP70 3′UTR that binds eEF1A1, we generated three deletion constructs of stem-loops according to the m-fold predicted structure (*Figure 4—figure supplement 1B*). The deletions ΔSL1 (Δ7–34) or ΔSL3 (Δ217–226, Δ246–255) did not reduce eEF1A1 binding to the 3′UTR significantly (*Figure 4—figure supplement 1B*), whereas the deletions in stem-loop 2 (ΔSL2–Δ83–93, Δ151–158) abolished eEF1A1 binding, indicating that this region was the primary interacting site for eEF1A1. The Kd for the eEF1A1-HSP70 3UTR complex was estimated to be ~100 nM (*Figure 4—figure supplement 1C*).

Since we observed eEF1A1 associated specifically with the HSP70 promoter and then with the RNAPII elongation complex, we examined if HSP70 mRNA stabilization by eEF1A1 required both the cognate promoter and the 3′UTR. To test this, we placed the luciferase ORF, with or without the HSP70 3′UTR, under regulation of the SV40 or HSP70 promoter. Upon heat shock, only the presence of both the HSP70 promoter and 3′UTR resulted in a steep decrease of luciferase activity in eEF1A1-knocked down cells (*Figure 4E*). In contrast, when HSP70 containing its 3′UTR was expressed from an SV40 promoter, no decay was evident. This supports the model that eEF1A1-mediated mRNA stabilization resulted from the recruitment of eEF1A1 during transcription and its subsequent association with the 3′UTR.

## eEF1A1 facilitates HSP70 mRNA nuclear export

It is well established that export of non-HSP RNAs is inhibited during HS (*Carmody et al., 2010*). We observed HSP70 mRNA in the cytoplasm of HS cells shortly after transcription activation (*Figure 3A*, *Figure 3—figure supplement 2*, second panel from the left [30 min of HS]). Since eEF1A1 has been reported to be a component of mammalian nuclear export machinery (*Khacho et al., 2008*), we investigated whether it plays a role in HSP70 mRNA transport. To gain further insight in the cellular

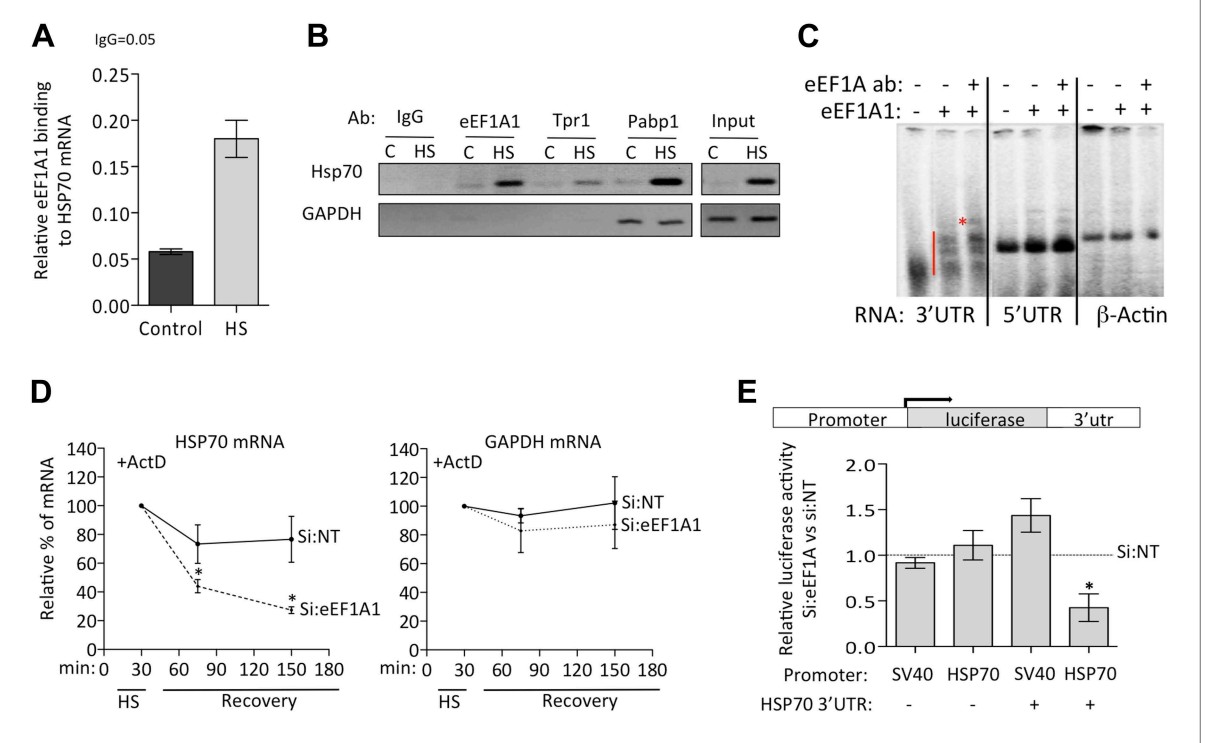

**Figure 4**. eEF1A1 binds the 3'UTR of HSP70 mRNA and stabilizes it. (**A**) eEF1A1 binds chromatin-associated HSP70 mRNA. Native ChIP samples were IPed using anti-eEF1A1 antibodies and reverse transcribed followed by HSP70 mRNA quantification. Data are represented as mean ± SEM from three independent experiments. (**B**) eEF1A1 interacts with the HSP70 mRNA in vivo as detected by RNA-IP. The panel shows the RT-PCR products of HSP70 and GAPDH mRNA from control (C) and heat-shocked (HS) cells after IP with indicated antibodies. IgG indicates the mock control. Input is total RNA. (**C**) Interaction between the HSP70 mRNA 3'UTR and eEF1A1 as detected by RNA-EMSA. 4 µg of purified eEF1A1 were incubated with 105 cpm of the 3'UTR or 5'UTR of HSP70 mRNA or a 200 nt fragment of the β-actin ORF radiolabeled by in vitro transcription. Only the 3'UTR of HSP70 mRNA was shifted by eEF1A1 (red line) and further super-shifted by 4 µg of antibody against eEF1A1 (red star). (**D**) Knock down of eEF1A1 diminishes HSP70 mRNA stability. Actinomycin D (Ac) was added 30 min after the onset of heat shock. The level of HSP70 and GAPDH mRNA at this time was taken as 100%. Data are represented as mean ± SEM from three experiments. *p < 0.05. (**E**) eEF1A1-mediates luciferase expression cloned in a HSP70 backbone plasmid. Mock transfected (si:NT) or eEF1A1-knocked down (si:EF) MDA-MB231 cells were transfected with plasmids expressing the SV40- or HSP70-driven luciferase gene fused to the HSP70 3'UTR, and a SV40-renilla luciferase plasmid used as a control. Luciferase activity was measured in cells heat-shocked for 1 hr at 43°C followed by 4 hr of recovery at 37°C. The values were normalized to those of renilla in the same cellular extracts. Bars represent luciferase activity in eEF1A1-deficient cells relative to that obtained for mock-transfected cells. Data are represented as mean ± SEM from three experiments. *p < 0.05.

The following figure supplement is available for figure 4:

**Figure supplement 1**. eEF1A1 binds the 3'UTR of HSP70 mRNA and stabilizes it.

localization of HSP70 mRNA, we used FISH in the immortalized MEFs expressing eEF1A1 fused to Cherry and Flag (***Figure 1—figure supplement 5A***). Cells knocked down for eEF1A1 showed a stronger FISH signal in the nucleus than non-transfected cells, where most of the HSP70 mRNA was localized in the cytoplasm (***Figure 5A***, ***Figure 5—figure supplement 1A,B***). Therefore, we quantified the effect of eEF1A1 in the partitioning of HSP70 mRNA between nucleus and cytosol after heat shock using RT-QPCR. After 2 hr of heat shock more than 95% of the total HSP70 mRNA has been exported to the cytoplasm of mock-transfected cells. In contrast, cells knocked down for eEF1A1 have accumulated ~30% of the total synthesized HSP70 mRNA in the nucleus (***Figure 5B***). We did not find any significant difference for GAPDH mRNA accumulation in the nucleus (~15% in mock transfected cells vs ~12% in cells knocked down for eEF1A1).

TPR1 is a nuclear pore complex (NPC) factor essential for HSP70 mRNA export (***Skaggs et al., 2007***). We postulated that eEF1A1 mediated the stress-induced binding of the HSP70 mRNA 3'UTR to TPR1. RNA-IP followed by RT-QPCR showed that eEF1A1 depletion greatly diminished the

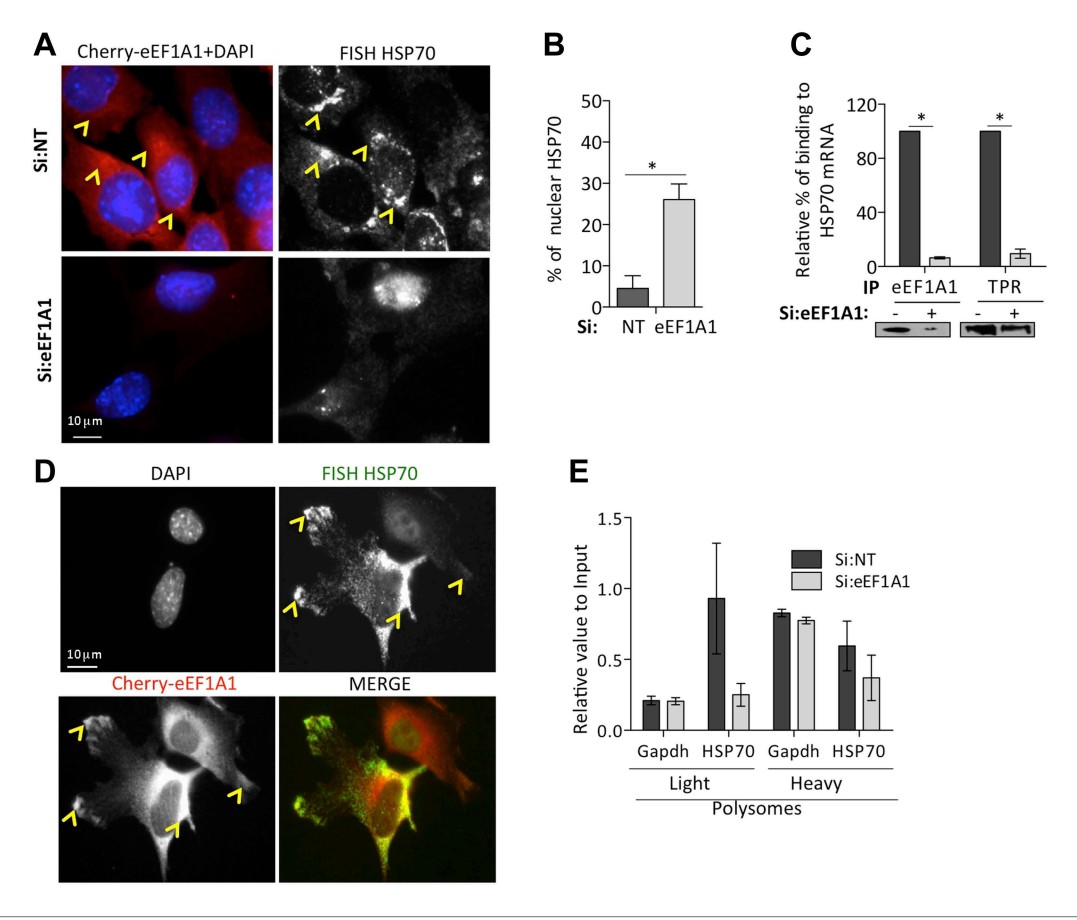

**Figure 5**. eEF1A1 is required for HSP70 mRNA export from the nucleus. (**A**) eEF1A1 mediates HSP70 mRNA export upon HS. MEFs were infected with a lentivirus expressing Cherry-eEF1A1. eEF1A1 expression was knocked down by siRNA. At 120 min after HS cells were fixed and HSP70 mRNA detected by FISH. Nucleus stained with DAPI. Merged images show cherry-eEF1A1 in red and nucleus in blue. Gray image shows HSP70 mRNA FISH. Bar = 10 microns. (**B**) HSP70 mRNA retention in the nucleus of eEF1A1-knocked down cells. The plot shows total (T) and nuclear (N) HSP70 mRNA in control (Si:NT) or eEF1A1-knocked down cells after heat shock. RNA from HeLa cells was RT with random primers followed by quantification of GAPDH and HSP70 mRNAs by QPCR. Total or nuclear HSP70 mRNA was normalized to that of total GAPDH. Data are represented as the mean ± SEM from three independent experiments. *p < 0.05. (**C**) Knock down of eEF1A1 suppresses binding of TPR1 to HSP70 mRNA. HSP70 mRNA was co-IPed with antibodies against eEF1A1 or TPR1 from heat-shocked HeLa cells knocked down of eEF1A1 or mock transfected. Total and IP RNA was RT with random primers and GAPDH and HSP70 mRNAs were quantified by QPCR. Total and IP HSP70 mRNA was normalized against total GAPDH mRNA. Data are represented as the mean ± SEM from three experiments. (**D**) MEFs were infected with a lentivirus expressing Cherry-eEF1A1. At 120 min of HS, cells were fixed and HSP70 mRNA detected by FISH. Nucleus stained with DAPI. Merged images show HSP70 FISH in green and cherry-eEF1A1 in red. Yellow arrowheads indicate areas with high density of HSP70 mRNA and brighter signal for cherry-eEF1A1. (**E**) eEF1A1 contributes to loading of HSP70 mRNA into polysomes. RNA collected from light and heavy polysome fractions was reverse-transcribed with random primers followed by quantification of GAPDH and HSP70 mRNAs by QPCR. Values relate to those obtained from total RNA (Input). Data are presented as the mean ± SEM from three independent experiments.

The following figure supplement is available for figure 5:

**Figure supplement 1**. eEF1A1 facilitates the export of HSP70 mRNA.

percentage of HSP70 mRNA bound to TPR1 upon heat shock (*Figure 5C*). Furthermore, both proteins co-immunoprecipitate (*Figure 5—figure supplement 1C*) during heat shock. This suggests a role for eEF1A1 in docking HSP70 mRNA to TPR1 in the NPC.

Once in the cytoplasm, HSP70 is selectively translated (*Cuesta et al., 2000*). We observed that the areas in the cytoplasm, both perinuclear and the leading edges, with higher density of HSP70 mRNAs corresponded to a brighter signal from cherry-eEF1A1 (*Figure 5A,D*, *Figure 5—figure supplement 1A*, [yellow arrows]). We did not observe such a high-density accumulation of HSP70 mRNAs in cells knocked down for eEF1A1 (*Figure 5A,D Figure 5—figure supplement 1A,B*). This could be explained by the decreased amount of HSP70 mRNA, but also by a less efficient loading of HSP70 mRNA into polysomes. Upon heat shock, the polysome profile MDA-MB231 cells knocked down for eEF1A1 was similar to that from mock transfected cells but, we detected a smaller amount of HSP70 mRNA in the pooled fraction of light ribosomes in cells knocked down for eEF1A1 (*Figure 5E*, *Figure 5—figure supplement 1C*), suggesting that eEF1A1 helped the loading of HSP70 mRNA into polysomes.

## Discussion

### eEF1A1 coordinates the heat shock response

HSR is a tightly regulated and synchronized process that is essential for cell survival under stress. While many HSP mRNAs, such as HSP70, are present in only very low amounts in unchallenged cells, their synthesis, stability, and translation amplify dramatically upon stress (*Theodorakis and Morimoto, 1987*; *Lindquist and Craig, 1988*; *Cuesta et al., 2000*; *Zhao et al., 2002*). We observed that each of these steps gets impaired when eEF1A1 is partially depleted. It is the sum of the effects on HSP70 mRNA synthesis, stability, and transport that accounts for a ~70% reduction of the HSP70 protein level and compromised thermotolerance in cells partially knocked down for eEF1A1. It is likely that the remaining eEF1A1 supports the residual HSP70 expression.

We have monitored the mRNA expression of the HSP70 by single molecule FISH in cells expressing Cherry-eEF1A1 (*Figures 3A, 5A,C*, *Figure 3—figure supplement 2*). We have found that although the expression of HSP70 during heat shock is synchronized, not all the cells respond with the same intensity at the same time. Indeed, induction of HSP70 transcription within the TSs of the same cell is stochastic (*Figure 3A*, *Figure 1—figure supplement 5B,C*, *Figure 3—figure supplement 2*). Nonetheless, most cells switched from almost undetectable levels of HSP70 mRNA under non-stress conditions to a massive induction of transcription from 30 to 60 min of heat shock, when mRNA can also be detected in the cytoplasm. Although the stimulus persisted, transcription ends at 2 hr of heat-shock. At this time point the majority of HSP70 mRNA is located in the cytoplasm. Our results suggest that eEF1A1 tags HSP70 mRNA during its transcription to ensure its rapid and efficient translation in contrast to non-HSP mRNAs (*Figure 6*). This hypothesis is in agreement with recent publications showing that the fate of the messenger is predetermined from transcription (*Harel-Sharvit et al., 2010*; *Bregman et al., 2011*; *Dahan et al., 2011*; *Trcek et al., 2011*; *Haimovich et al., 2013*).

eEF1A1 delivers aminoacylated tRNAs to the A site of ribosome during translation elongation (*Li et al., 2013*). We found that upon HS, when translation elongation is paused to protect cellular homeostasis (*Slobin, 1980*; *Morimoto et al., 1997*), eEF1A1 accumulates in discrete dots in the nucleus that correspond to HSP70 TS (*Figure 3A*, *Figure 3—figure supplement 2*). eEF1A1 participates in the regulation of transcription by recruiting and activating HSF1 (*Figure 2A,B*). Formation of the HSF1-eEF1A1-DNA complex in vitro and in vivo requires an RNA component (*Figure 2—figure supplement 1B*), which is thought to function as a thermosensor (*Shamovsky et al., 2006*). eEF1A1 also interacts with the hyperphosphorylated form of RNAPII upon heat shock (*Figure 3E*), suggesting that it travels with RNAPII during elongation of HSP mRNA (*Figure 3D*, *Figure 3—figure supplement 4*). Although it is possible that binding of eEF1A1 to HSP70 mRNA can be independent of transcription elongation, the association of eEF1A1 with RNAPII (*Figure 3E*) may facilitate relocation of eEF1A1 from the promoter to the 3′UTR of HSP mRNA (*Figure 3D*, *Figure 3—figure supplement 4D*) and thus may be crucial for the stability and transport of HSP mRNAs (*Figures 4 and 5*).

The stability of HSP70 mRNA sharply increases upon stress via a highly conserved mechanism involving *cis*-acting AU rich elements (ARE) in the 3′UTR (*Theodorakis and Morimoto, 1987*; *Zhao et al., 2002*). Heat shock inhibits ARE-facilitated mRNA decay by stabilizing HSP70-AUF1 (ARE/poly(U)-binding/degradation factor 1) and PABP1 binding to the 3′UTR (*Laroia et al., 1999*; *Wang and Kiledjian, 2000*). Also, PKR (double-stranded RNA-dependent protein kinase) plays a significant role in stabilizing HSP70 mRNA upon stress (*Zhao et al., 2002*). eEF1A1 binds directly to the 3′UTR of HSP70 mRNA during heat shock (*Figure 4C*, *Figure 4—figure supplement 1B*). Knock down

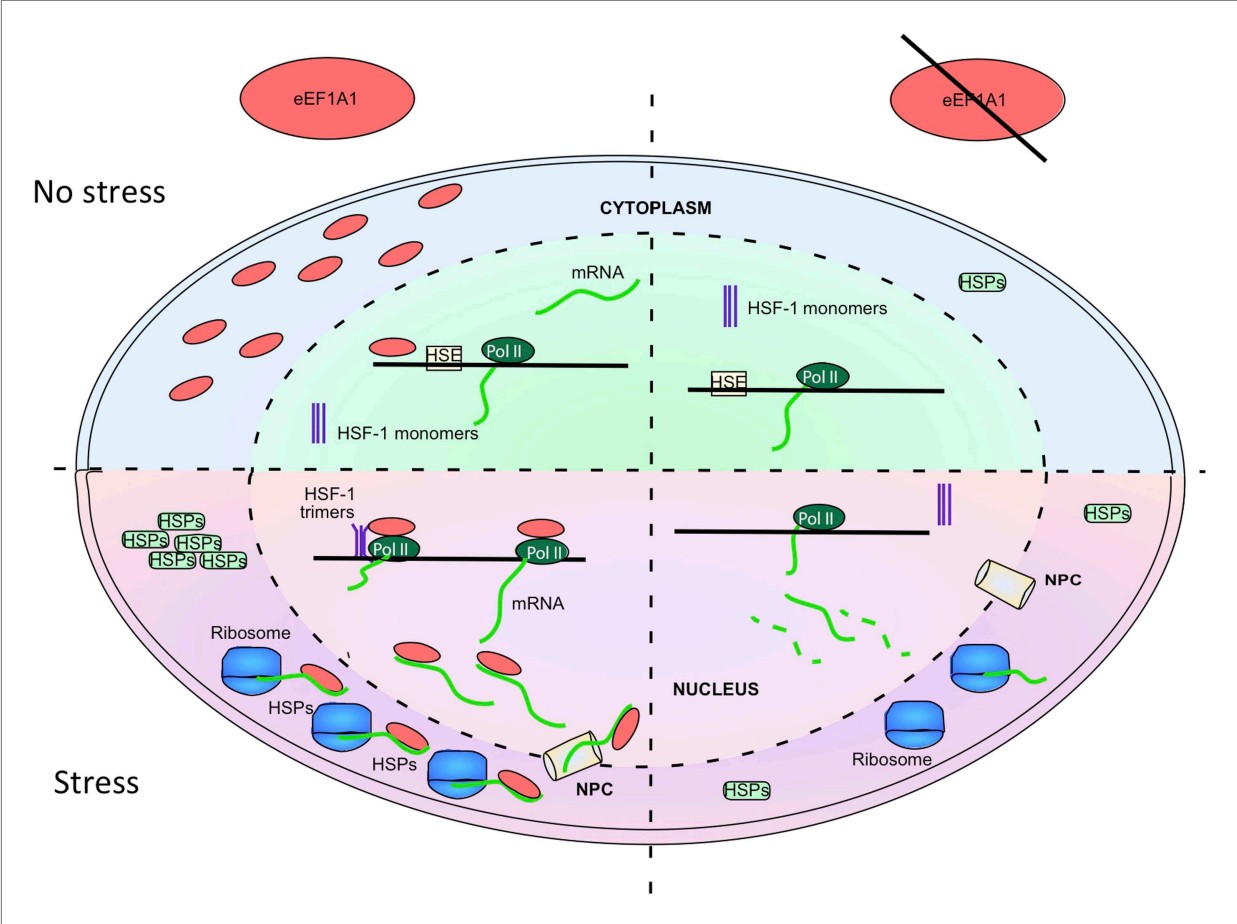

**Figure 6**. eEF1A1 synchronizes the expression of HSP70 mRNA from transcription to translation. The cartoon summarizes the results of this study, implicating eEF1A1 at each stage of HSPs induction. Prior to stress, eEF1A1 resides mostly in the cytoplasm where it is an essential component of the translation machinery (top left quadrant). Upon heat shock (bottom left quadrant), a fraction of eEF1A1 is detected in the HSP70 locus where it recruits HSF1 to an HSP70 promoter, thus activating transcription. eEF1A1 interacts with elongating RNAP II and binds the 3'UTR of HSP70 mRNA to stabilize the transcript and to export it to cytoplasm for efficient translation. By synchronizing all the major steps of HSP70 gene expression, eEF1A1 renders the process of HSR exceptionally robust and coordinated. In cells knocked down for eEF1A1 HSP70 remain at low levels even when cells are stressed (right hand quadrants).

of eEF1A1 results in selective destabilization of HSP70 mRNA (*Figure 4D*, *figure 4—figure supplement 1B*). The interaction of eEF1A1 with *trans*-acting factors, such as AUF1 and PABP1, as well as the phosphorylation and/or methylation status of eEF1A1 in response to different stimuli is likely to determine the stability of specific mRNAs and the interaction of eEF1A1 with HSP70 mRNA. It is also possible that PKR influences the stability of HSP70 mRNA (*Zhao et al., 2002*) via phosphorylation of eEF1A1 (*Xue et al., 2008*).

Once HSP mRNAs have been properly processed they must be promptly exported from the nucleus to ribosomes in the cytoplasm. HSF1-mediated recruitment of the NPC protein TPR1 stimulates export of HSP70 mRNA from the nucleus (*Skaggs et al., 2007*). Inhibition of the HSF1–TPR1 interaction suppresses HSP70 mRNA export without decreasing mRNA stability (*Skaggs et al., 2007*). Since eEF1A1 interacts with TPR1 during heat shock (*Figure 5—figure supplement 1C*) and is required for TPR1 binding to HSP70 mRNA (*Figure 5C*) and export of HSP70 mRNA from the nucleus (*Figure 5A,B*), the transport mechanism appears to rely on eEF1A1-mediated activation of HSF1 and formation of the ternary complex of TPR1-HSF1-eEF1A1 with HSP mRNA.

Decreased transport correlated with poor loading of HSP mRNAs into active ribosomes (*Figure 5E*, *Figure 5—figure supplement 1D*), whereas both, polysome profiles and general protein synthesis, were unaffected in cells knocked down for eEF1A1 (*Figure 1A—figure supplement 1C*), suggesting

a selective effect of eEF1A1 on HSP70 translation. It is likely that eEF1A1 modulates the expression of many genes controlled by HSF1. Our results with HSP27 and other HSPs support this hypothesis (*Figure 1—figure supplement 2C,D,F*, *Figure 5—figure supplement 1D*).

## Clinical implications

Deregulation of HSR is associated with numerous pathologies ranging from cancer to neurodegenerative conditions. A large effort has been made to develop pharmacological approaches to either inhibit or stimulate HSR to, respectively, kill rogue cells or protect normal cells (*Calderwood et al., 2006*; *Demidenko et al., 2006*; *Galluzzi et al., 2009*; *Whitesell and Lindquist, 2009*; *Neef et al., 2011*). The induction of HSP70 in transgenic mouse models of spinal bulbar muscular atrophy (SBMA) and amyotrophic lateral sclerosis (ALS) ameliorated disease symptoms and prolonged lifespan (*Bruening et al., 1999*; *Adachi et al., 2003*). The present findings illuminate eEF1A1 as a promising tool for the treatment of these diseases. eEF1A has two isoforms, 1 and 2, expressed from two different genes that are 96% similar at the protein level. In contrast to most other cell types, mature motor neurons express only eEF1A2 instead of eEF1A1 (*Figure 1—figure supplement 3C*; *Newbery et al., 2007*). Because eEF1A2 does not support HSR (*Figure 1—figure supplement 3A,B*), our results provide an explanation of why motor neurons do not induce HSPs upon stress, even though they express high levels of HSF1 (*Batulan et al., 2003*). Our findings also suggest a novel therapeutic strategy for these neurodegenerative diseases by enabling eEF1A1 expression in motor neurons.

In summary, the isoform 1 of translation elongation factor eEF1A has been established as a key component of HSR that controls each stage of the HSP70 induction from transcription activation to mRNA stabilization, nuclear transport, and translation (*Figure 6*). This remarkable feature of eEF1A1 makes it an attractive target for the treatment of many pathological conditions associated with HSR deregulation.

## Materials and methods

### Cell culture, treatments, and transfections

293T, TIGs, MDA-MB231, and HeLa cells were obtained from ATCC (ATCC, Manassas, VA, USA); NSC34 cells were obtained from Cedarlane (Cederlane, Burlington, NC, USA). MEFs from the homozygous *Actb*-MBS mouse strain have been previously described (*Lionnet et al., 2011*). All cell lines were cultured in DMEM supplemented with 5% FBS and P/S at 37°C in an atmosphere of 5% $CO_2$.

To induce HSR, cells grown in tissue-culture flasks were submerged in a water bath at 42°C or 43°C (for cells of human or mouse origin respectively) for the indicated time intervals. Translation was inhibited by cycloheximide (CHX, 50 µg/ml) or doxycycline (DOX, 5 µg/ml) (Sigma, Saint Louis, MO, USA). Arsenite stress was induced with 0.1 mM arsenite (Sigma, Saint Louis, MO, USA). Transcription was inhibited with 5 µg/ml of actinomycin D (Sigma, Saint Louis, MO, USA) or 100 µM of DRB (Cayman, Ann Harbor, Michigan, USA).

DsiRNA for eEF1A1 (NM_001402.5–*Homo sapiens*, NM_010106.2–*Mus musculus*), eEF1A2 (NM_001958.2–*H. sapiens*), and HSF1 (NM_005526.2–*H. sapiens*) were designed by IDT (IDT, Coralville, Iowa, USA) (http://www.idtdna.com/catalog/DicerSubstrate/Page1.aspx). Two DsiRNAs were selected for each gene: NM_001402.5 (1 and 10 [pair A] and 3 and 5 [pair B]) (*Yan et al., 2008*) NM_010106.2 (1 and 6 [pair A] and 3 and 5 [pair B]) NM_005526.2 (1 and 2) and NM_001958.2 (1 and 2). Inhibition of the desired gene was obtained 48 hr after the second co-transfection of each DsiRNA at a final concentration of 0.6 nM with the recommended amount of trifectin transfection reagent according to the manufacturer's instructions (or 10 nM of each siRNA transfected with SilenFect [Biorad, UK])). Consecutive co-transfections were performed at 48 hr intervals in antibiotic free media. DS scrambled negative (IDT), which did not target any gene (Si:NT), was used as a negative control.

Transfection of expression plasmids was performed with lipofectamine 2000 (Invitrogen Life Technologies, Van Allen Way, Carlsbad, CA, USA) according to the manufacturer's instructions.

### Plasmid constructions

Renilla and firefly luciferase expression plasmid were pGL-3 control (Promega, Madison, WI, USA). SV40 promoter on Luciferase-pGL3 was substituted by HSP70 promoter fragment obtained after digestion of p1730R (Enzo Life Sciences, Farmigdale, NY, USA) with XhoI and HindIII. The HSP70 (HSPA1A) (NM_005345.2) 3′UTR sequence was obtained by PCR with specific primers (*Supplementary file 1A*), from total HeLa cell DNA, cloned in pJet 1.2/blunt (Fermentas Thermosicentific, Pittburg, PA, USA) and re-cloned with XbaI in pGL3-control.

eEF1A1 cDNA was obtained by RT, followed by PCR with specific primers (*Supplementary file 1A*), from total HeLa cell RNA. eEF1A1 cDNA was cloned with BamHI and ClaI into pHAGE-Ubc-RIG lenti-viral vector (a gift from Gustavo Mostolslavsky) to achieve the expression of eEF1A fused to cherry and Flag in its N terminus (phage-ubc-cherry-flag-eEF1A1).

## Lentivirus production and cell sorting

30 µg of Phage-ubc-cherry-flag-eEF1A1 was transfected together with rev, gag/pol, tat and vsg plasmid using lipofectamine2000 (Life Technologies, Van Allen Way, Carlsbad, CA, USA) in 293T cells growing at 80% of confluence. Lentivirus were collected every 24 hr for 3 days after transfection, combined, and concentrated to 1 ml with lenti-X concentrator (Clontech, Mountain View, CA, USA). $2 \times 10^6$ MEFs or TIGs were infected with 300 ml of lentivirus and 4 days after infection cells were sorted for cherry expression in the flow cytometry core facility at the Albert Einstein College of Medicine.

## In vitro transcription

In vitro transcription and labeling of the 5′UTR and 3′UTR of HSP70 (NM_005345.2) mRNA, and a 200nt region of the ORF of β-actin (NM_00,101.3) RNA was performed with [$^{32}$P] CTP (MP Biomedicals, Santa Ana, CA, USA) and the MEGAscript T7 kit (Ambion, Grand Island, NY, USA) from PCR products (*Supplementary file 1A*). PCR was performed using cDNA templates previously cloned in the pJem1.2/blunt vector from CloneJet PCR Kit (Fermentas Thermosicentific, Pittburg, PA, USA). PCR conditions for Takara polymerase were 32 × (95:15″, 60:15″, and 72:45″). PCR products were purified with the minElute PCR purification kit (Qiagen, Valencia, CA, USA). The sequences of all cDNA products were verified by sequencing (Macrogen, New York, NY, USA) from the T7 promoter (pJem1.2, Fermentas Thermosicentific, Pittburg, PA, USA) and internal regions using specific primers.

## RNA EMSA and EMSA

Specific RNA probes for EMSA were synthesized in vitro and radiolabeled as described above. eEF1A1 protein was purified as follows: 50 ml of the HeLa S-100 lysate was applied to a 80-ml Q Sepharose column whose outlet was connected to a 40-ml CM Sepharose column. Both columns were equilibrated with 7.5/0 buffer (50 mM Tris–HCl, pH 7.5, 0.5 mM EDTA, 2 mM DTT, 10% glycerol) at a flow rate of 1 ml/min. The column was washed with 7.5/0 buffer until most of the unbound protein appeared in the flow-through (as determined at $A_{280}$). The Q Sepharose column was then disconnected, and the CM Sepharose column was developed with 10 column volumes of a 50–650 mM gradient of KCl in 7.5/0 buffer at a flow rate of 1 ml/min, while fractions of 4 ml were collected. Fractions containing eEF1A were identified by immunoblotting, pooled, dialyzed against buffer containing 50 mM Tris–HCl, pH 7.5, 100 mM KCl, 0.5 mM EDTA, 2 mM DTT, 10% glycerol, and concentrated with Millipore Centrifugal Filter units (3 kDa MWCO). The concentrated protein was flash-frozen in liquid nitrogen and stored at −80°C. RNA EMSA was accomplished according to the protocol described by *Yan et al. (2008)*. Briefly, RNA probes were in vitro transcribed, end labeled, and gel purified. Indicated concentrations of purified eEF1A1 were incubated with 20 kcpm of HSPA1A 3′UTR RNA (~0.1 pmol/µl) in the presence of 50 µM GTP for 30 min at room temperature.

EMSA and RNAse treatments were performed as described previously (*Shamovsky et al., 2006*; *Yan et al., 2008*). For supershift experiments, 10 µg of cell extracts were incubated with 1 µg of the specific antibody (HSF1 and HSF2 [Enzo Life Sciences, Farmigdale, NY, USA], eEF1A [Millipore, Billerica, MA, USA], IgG [Abcam, Cambridge, MA, USA]) for 1 hr prior to incubation with radiolabel HSE.

## RT, PCR, and QPCR

Total cell RNA was extracted with the RNAeasy mini kit (Qiagen, Valencia, CA, USA). 2 µg of RNA were treated with turbo DNAse (Ambion, Grand Island, NY, USA) and reverse transcribed with random primers or oligo dT using MLV-RT (Promega, Madison, WI, USA). 5 µl of a 1:15 dilution of cDNA were used for QPCR with specific primers (*Supplementary file 1A*) and Power SYBR Green PCR master mix 2× (Applied Biosystems, Forest City, CA, USA) for 40 cycles in a 7300 real-time PCR system (Applied Biosystems, Forest City, CA, USA) according to the manufacturer's instructions. HSP70 *Ct* was normalized to GAPDH *Ct* for each condition and this value was related to the control value.

Takara polymerase (TaKara, Moonachie, NJ, USA) was used for PCR following the instructions of the manufacturer.

For ChIP experiments Real-time QPCR was performed in a Stratagene Mx3005p with Brilliant II SYBR Green kits (Stratagene, Netherlands) according to the manufacturer's instructions ans specific primers (*Supplementary file 1A*). Data were computed as described (*Saint-Andre et al., 2011*).

## Polysome gradients and RNA extraction

Mock-transfected or eEF1A1 knocked down MDA-MB-231 cells were heat shocked for 45 min at 43°C and allowed to recover for 45 min at 37°C. At this time cells were treated with 100 µg/ml of cycloheximide for 15 min and collected for polysome purification using the protocol, centrifuge and ISCO fraction collector described by *Ramirez-Valle et al. (2008)* without modifications. Total RNA or RNA collected from polysome fractions was reverse transcribed and quantified as described above.

## Metabolic labeling

Cells were labeled with 50 µCi of [$^{35}$S]-methionine per ml (Easytag Express Protein Labeling Mix, Dupont/NEN) as described (*Cuesta et al., 2000*) for both control and heat-shock conditions.

## Cell viability and death

Cell viability was measured by the MTT assay (Promega), and the OD was measured in an Infinite M200 96 well plate reader (Tecan) 24 hr after the second round of siRNA transfection. Cell death was quantified by flow cytometry (Becton Dickinson FACScalibur) after cells were stained with propidium iodide buffer (PI) (Life Technologies, Van Allen Way Carlsbad, CA, USA). Data were analyzed with Sumit software.

## Immunoblotting

Cells were washed twice in 1× PBS, snap-frozen in liquid nitrogen and resuspended in RIPA buffer (50 mM Tris–HCl, pH 7.5, 1% NP-40, 0.5% sodium deoxycholate, 0.1% SDS, 1 mM EDTA, 150 mM NaCl, 1× protease inhibitor cocktail [Roche, Bransburg, NJ, USA] and 1× phosphostop [Roche, Bransburg, NJ, USA]) for 10 min at 4°C. 10 µg of protein were resolved on 4–12% SDS PAGE (Life Technologies, Van Allen Way Carlsbad, CA, USA) and transferred to a Nitrobound nitrocellulose membrane 0.45-µm pore size (Fisher). The membrane was blocked in 1× PBS-0.05% Tween 20 and 5% nonfat dry milk for 1 hr at room temperature and then incubated overnight at 4°C with specific antibodies (HSP70, HSF1, HSP27 (Enzo Life Sciences, Farmigdale, NY, USA), eEF1A1 (Millipore, Billerica, MA, USA), eEF1A2 (a gift from Jonathan Lee (*Khacho et al., 2008*) and GAPDH (Sigma, Sigma, Grand Island, NY, USA) and RNAPII (Santa Cruz, Dallas, Texas, USA)). After three washes in 1× PBS—0.05% Tween 20 membranes were incubated with horseradish peroxidase-conjugated goat-anti-mouse or goat-anti-rabbit antibody (GE-Healthcare, GE-Heathcare, Ho-Ho-Kus, NJ, USA), washed three times, and exposed to Kodak films using the ECL chemiluminescence system (GE-Heathcare, Ho-Ho-Kus, NJ, USA). Alternatively, IRD-ye and VRD-ye secondary antibodies were used before detection and quantification with the Odyssey infrared imaging system (LI-COR bioscience, Lincoln, Nebraska, USA).

## Co-immunoprecipitation (co-IP)

Cells were lysed in nonidet lysis buffer (25 mM Hepes, pH 7.9, 100 mM NaCl, 5 mM EDTA, 0.5% nonidet P40, 1× complete [Roche, Bransburg, NJ, USA], and phosphatase inhibitors [Roche, Bransburg, NJ, USA]) for 1 hr at 4°C. Lysates were clarified of non-protein components by centrifugation for 5 min at 14,000×*g* and pre-cleared with 25 µl of Dynabeads protein G (Life Technologies, Van Allen Way Carlsbad, CA, USA) washed three times with B150 buffer (20 mM Tris, pH7.4, 1 mM EDTA, 10% glycerol, 150 mM NaCl and 0.1% Triton) for 1 hr at 4°C. 1 mg of pre-cleared protein was IP overnight at 4°C with 5 µg of anti-eEF1A1 (Millipore, Billerica, MA, USA) or normal rabbit or mouse IgG (Abcam, Cambridge, MA, USA) followed by a 1-hr incubation with 50 µl of Dynabeads protein G (Life Technologies, Van Allen Way Carlsbad, CA, USA). After five washes in B150, the proteins were resolved by SDS-PAGE and analyzed by immunoblotting.

## RNA-IP, Chromatin-IP (ChIP), and Native RNA-ChIP

RNA-IP experiments were performed as described previously (*Skaggs et al., 2007*; *Saint-Andre et al., 2011*). For each RNA-IP 5 µg of specific antibody: eEF1A (Millipore, Billerica, MA, USA), TPR1 (Santa Cruz Dallas, Texas, USA), PABP (Abcam, Cambridge, MA, USA), or IgG (Abcam, Cambridge, MA, USA) was used. Samples were analyzed by RT-QPCR or RT-PCR with specific primers (*Supplementary file 1A*).

ChIP analysis were performed as described by *Saint-Andre et al. (2011)* Antibodies were: rabbit or mouse IgG (Sigma, Grand Island, NY, USA), RNAPII (2 µg of Antibody for 20 µg of relative amount of

chromatin A260) (Santa Cruz, Dallas, Texas, USA), RNAPII S2 (2 µg of Antibody for 20 µg of relative amount of chromatin A260) (Abcam, Cambridge, MA, USA), eEF1A (4 µg of Antibody for 80 µg of relative amount of chromatin A260) (Designed in Nudler lab and produced in rabbits and purified by Prosci incorporated), HSF1 (2 µg of Antibody for 20 µg of relative amount of chromatin A260) (Enzo life sciences, Farmigdale, NY, USA), or Flag (2 µg of Antibody for 80 µg of relative amount of chromatin A260) (Sigma, Sigma, Grand Island, NY, USA). Pellets were resuspended in 40 µl of water and 1 µl analyzed by QPCR with specific primers (*Supplementary file 1A*). Percentage input for each data point was calculated by comparing the *Ct* value of sample to a standard curve generated from *Ct* of a 5-point serial dilution of input chromatin. Each immunoprecipitation was performed and assayed at least three times from independent samples.

## Nuclear run on

Transcriptional run-on assays were performed as described in *Banerji et al. (1984)* using nuclei prepared from primary fibroblasts knocked down of eEF1A1 and heat shocked for 20 min at 43°C; radioactively α-$^{32}$P-labeled RNA was prepared and hybridized to single-stranded oligos immobilized on nitrocellulose filters and visualized by PhosphorImager, quantitated by ImageJ.

## RNA-FISH

We performed RNA FISH on cultured cells to a protocol modified from previously published protocols (*Lionnet et al., 2011*). A set of Stellaris FISH probes single-labeled (quasar 670) oligonucleotides were designed to selectively bind to human HSPA1A or mouse HSPA1A (*Supplementary file 1B*) (Biosearch technologies, Petaluma, CA, USA) or MS2 (*Lionnet et al., 2011*).

We grew MEFs on coverslips in DMEM 10% FBS 1% pen-strep, then fixed them in 4% paraformaldehyde for 15 min at room temperature before washing and storing in PBSM (PBS supplemented with 5 mM MgCl2) at 4°C. Before hybridization, we permeabilized the cells 10 min in 0.5% triton X-100 in PBS, then washed them in PBS 10 min and incubated 60 min at 37°C in prehybridization solution (20% formamide, 2× SSC, 2 mg ml⁻¹ BSA, and 200 mM vanadyl ribonucleoside complex). We then hybridized the probes to the cells for 4 hr in hybridization solution (10% dextran sulfate, 2× SSC, 10% formamide, 2 mg ml⁻¹ BSA, 0.2 mg/ml *E. coli* tRNA, 200 mM vanadyl ribonucleoside complex, and 2U of RNAsin) supplemented with 10 ng DNA probe (for MS2) or 1.25 mM (for human and mouse HSPA1A) per coverslip. We washed coverslips twice 20 min at 37°C with prehybridization solution, then 10 min at room temperature in 2× SSC, and 10 min at room temperature in PBSM. We counterstained DNA with DAPI (0.5 mg l⁻¹ in PBS). After a final wash in PBS, we mounted coverslips mounted on slides using ProLong gold reagent (Life Technologies, Van Allen Way Carlsbad, CA, USA).

## Image acquisition and analysis

Images were acquired on an Olympus BX61 epifluorescence microscope with PlanApo 60×, 1.4NA, and UPlanApo 100×, 1.35NA, oil-immersion objectives (Olympus). An X-Cite 120 PC (Lumen Dynamics, Canada) light source was used for illumination, with filter sets 31,000 (DAPI, CMAC), 41007a (Cy3) and 41,008 (Cy5) (Chroma Technology, Bellows Falls, VT, USA). We acquired data using 21 optical sections with a z-step size of 0.2 µm using a CoolSNAP HQ camera (Photometrics) with a 6.4 µm-pixel size CCD. MetaMorph (Molecular Devices) software was used for instrument control as well as image acquisition.

We automatically quantified the position and intensity of transcription sites within the nucleus of MEFs using software described in detail in *Lionnet et al. (2011)*. Briefly, the two-part algorithm begins by identifying the approximate position of transcription sites by applying a spatial bandpass filter to the image. The software identifies clustered pixels above a user-determined threshold as transcription sites. Then, the software fits a two-dimensional Gaussian, which approximates a point spread function, to determine the spatial position and intensity of each transcription site, after subtracting a local estimate of the fluorescent background. After the center position of TS spots in each channel were identified in 3-dimensional space, colocalization events were assigned when the distances between the closest spot of different colors were within two pixels.

## Statistical analysis

Values were expressed as standard error mean (SEM). Statistical significance was assessed by paired t test or ANOVA test.

## Acknowledgements

We thank Gustavo Mostoslavsky, Jonathan Lee, and Ilya Shamovsky for materials. We also thank Montse Cols, Helen J Newbery, Jemmy Cheng, and Singerlab members (Carolina Eliscovish, Bin Wu, Xiuhua Meng, Melissa Lopez-Jones, and Shailesh M Shenoy) for technical assistance and Danny Reinberg, Rafael Cuesta-Sanchez, Maria del Valle and Emilio Lecona for discussions and comments. We thank Timur Artemyev for his continuous support. We thank AECOM NCI (P30CA013330) for the cell sorting at the flow cytometry facility at Einstein. This work was supported by the Fundacion Ramon Areces and Fondation ARSEP (MV), Wellcome Trust (CMA), NIH grants R01GM57071 (RS) and R01 AI090110 (EN), Robertson Foundation and Howard Hughes Medical Institute (EN).

## Additional information

### Competing interests

RHS: Reviewing editor, *eLife*. The other authors declare that no competing interests exist.

### Funding

| Funder | Author |
| --- | --- |
| Howard Hughes Medical Institute | Evgeny Nudler |
| National Institutes of Health | Maria Vera, Bibhusita Pani |

The funders had no role in study design, data collection and interpretation, or the decision to submit the work for publication.

### Author contributions

MV, Conception and design, Acquisition of data, Analysis and interpretation of data, Drafting or revising the article; BP, LAG, Acquisition of data, Analysis and interpretation of data; CM, Analysis and interpretation of data, Contributed unpublished essential data or reagents; CMA, Acquisition of data, Analysis and interpretation of data, Contributed unpublished essential data or reagents; RHS, Analysis and interpretation of data, Drafting or revising the article; EN, Conception and design, Analysis and interpretation of data, Drafting or revising the article

## Additional files

### Supplementary file

• Supplementary file 1. (**A**) Primers used in this work. (**B**) Fish probes used in this work.

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
