## [Decision Letter]

Thank you for choosing to send your work entitled “eEF1A1 couples HSP70 transcription to translation during heat shock response” for consideration at *eLife*. Your full submission has been evaluated by James Manley (Senior editor) and 2 peer reviewers, one of whom is a member of our Board of Reviewing Editors, and the decision was reached after discussions between the reviewers. As you will see in the reviews below, both reviewers raised a number of substantive issues that we feel precludes publication of the current version of your manuscript. Revisions for *eLife* papers are intended to be performed within 2-3 months and we believe that the amount of experimentation required to fully address the reviewers' comment will take more than that amount of time. However, *eLife* would be willing to consider a re-submitted manuscript that fully addresses the reviewers' comments.

Reviewer #:

In this manuscript Vera et al. report a new role for the translation elongation factor eEF1A in the heat shock response. They find that eEF1A acts at multiple steps in expression of heat shock proteins. First, they show that eEF1A binds to heat shock genes, such as HSP70, and helps recruit the heat shock transcription factor HSF1 to the promoter. They further show that eEF1A is associated with RNA POL II and the 3' UTR of HSP70 mRNA, stabilizing and facilitating it for nuclear transport and ultimately to active ribosomes.

The following issues should be addressed prior to publication.

1) In RNAi experiments such as those performed in this manuscript, it is essential to show that similar results are obtained using two unrelated siRNAs (or shRNAs) against the same target gene to rule out off-target effects. Ideally, this should be done for all experiments in the manuscript, but minimally this needs to be done for the key experiments in the manuscript from which the main conclusions are drawn.

2) Figure 1. It would be nice to see the results minus and plus heat shock.

3) Figure 2. Have the authors performed these in vitro DNA binding experiments with purified HSF and eEF1A (Figure 2 uses only purified eEF1A).

4) Figure 2. Isn't the fold-increase following heat-shock expected to be larger? Is there a comparable experiment for HSF1?

5) Figure 3. Why isn't time zero shown?

6) Figure 3. Isn't this experiment better suited to Figure 1. Why does DRB cause decreased occupancy of RNA pol II and eEF1A at the promoter?

7) Figure 4. What does relative on the Y-axis mean?

8) Figure 4. Has HSF been tested in the RNA-IP assay.

9) Figure 4. It would be nice to see the eEF1A binding site in the 3' UTR further localized.

10) Figure 4. It is not clear why the promoter is needed here. Doesn't the in vitro RNA binding data of Figure 4 show that eEF1A can recognize and bind directly to the 3' UTR?

11) For several of the panels (Figures 3, 4 and 5) statistics need to be provided.

Reviewer #2:

The manuscript by Vera et al claims the translation factor eEF1A participates at multiple steps in the human and mouse heat shock response (HSR): recruiting the HSF1 regulator to the promoter, associating with elongating Pol II and the 3'UTR of HSp70 mRNA, stabilizing the mRNA, and facilitating its export from the nucleus and to active ribosomes. These extraordinary claims, which in effect state that, “eEF1A orchestrates the entire heat shock response”, require extraordinary evidence. The authors provide a lot of experimental results in their effort to support all of these eEF1A1 functions. Some experiments are well-controlled and show reasonably convincing effects. Others are much less convincing.

How can a single 50 kDa factor do all of this? I am concerned that other general mechanisms, which do not require a direct mechanistic role in each of these very specific steps, might account for the numerous effects seen. Could the depletion of such a super abundant protein (1-3% of total cytosolic proteins) lead to general pleiotropic effects that make general cellular mechanisms sluggish? eEF1A1 is also known to have chaperone function (see J. Biol. Chem. 2007, 282:4076), and its depletion might have a broad array of effects including those seen here. The previously documented role of eEF1A1 by this group as having a role in trimerization and activation of HSF1 could be consistent with such a chaperone role. These general concerns and the lack of a mapping of specific functions to distinct parts or mutations of the protein, lead me to the view that this grand view of eEF1A is too ambitious for a single paper and for the data in hand. This is already a bulky paper and it may be better suited being published as a couple of more focused papers in more specialized journals than *eLife*. However, publication either in *eLife* or elsewhere requires addressing these general concerns and specific concerns below:

Specific concerns:

1) The claim that translation is not affected by knockdown of eEF1A1 is not adequately supported by the data. The authors show that total translation is unaffected in cells where eEF1A1 was reduced to ∼35% of normal levels (Figure 1—figure supplement 1). The heat shock for this experiment was done at 40C instead of 43C. All the other experiments were done at full heat shock temperatures 43C or 45C. The 40C appears not to be a typo, because, under their 40C heat shock, there is not the change in protein patterns expected for a normal heat shock. A broad effect on general translation in full heat shock conditions caused by.eEF1A1 could lead to reduced HSPs.

2) With no standard curve to show their Western assays are linear, it is inappropriate to ascribe the accuracy implied by a statement “knock down of eEF1A1 <38%” and in other numbers in Figure 1 and supplemental figures. It's not sufficient to normalize to a GAPDH band.

3) Figure 1 needs more controls. Is the complete loss of the red eEF1A only in the cytoplasm due to a 65% knock down of this protein??

4) Are there no error bars associated with the control in Figure 1—figure supplement 2 and Figure 1—figure supplement 3? If there are bars in proportion to the measurement of eEF1A1, then are the differences as significant as the authors’ state?

5) With eEF1A1 being so abundant, can many of the interactions observed with other proteins and RNAs in the cell be non-specific or low affinity (Figure 2). Perhaps more quantification and use of other proteins as controls that are similarly abundant would help.

6) In the binding of eEF1A1 to Hsp70 RNA, a single high concentration of protein is used. A binding curve and estimate of the Kd should be provided, if this is claimed to be an RNA binder.

7) It was recently reported that inhibition of translation inactivates Hsf1 (Santagata et al. Science, 2013). If the translation is found to be compromised in eEF1A1 knockdown cells, then many observations reported in this manuscript can be explained by the possibility of unaccounted effect of eEF1A1 knockdown in inactivation of Hsf1 and hence, attenuation of Hsp70 and other heat shock proteins expression upon heat shock.

8) Finally, is the proposed role of eEF1A1 in recruitment and activation of Hsf1 limited to major Heat Shock Protein genes or is it much broader and applicable for all Hsf1 regulated genes? Evidence exists for Hsf1 regulating expression of genes other than HSPs (Trinklein et al., 2004 & Mendilli et al., 2012) and I think it would be quite informative, if authors could include few of these genes and examine the role of eEF1A1 knockdown in their induction upon heat shock.

[Editors’ note: what now follows is the decision letter after the authors submitted for further consideration.]

Thank you for resubmitting your work entitled “eEF1A1 couples HSP70 transcription to translation during heat shock response” for further consideration at *eLife.* Your revised article has been favorably evaluated by James Manley (Senior editor), a member of the Board of Reviewing Editors, and an outside expert. The manuscript has been improved but there are some remaining issues that need to be addressed before acceptance, as outlined below.

The authors have addressed or clarified many of the concerns of the reviewers. The remaining concerns need to be addressed, but they can probably be dealt with by textual revisions.

1) The previous Impact Statement stated “eEF1A1 orchestrates the entire process of the heat shock response, from transcription activation of HSP70 gene, to HSP70 mRNA stabilization, nuclear export, and translation”. I apologize for abbreviating this to “eEF1A orchestrates the entire heat shock response”. The current Impact Statement now is “eEF1A1 orchestrates the process of heat shock response, from transcription activation of the HSP70 gene, to HSP70 mRNA stabilization, nuclear export, and translation”. The word “entire” is now deleted, which helps, but this is still an overstatement, especially considering the complexity of the heat shock response. First, the “process of the heat shock response” is too broad, as this study focuses on Hsp70. Second, the work “orchestrates” gives eEF1A1 a status of a major regulator like HSF1. While the paper provides data supporting an eEF1A1 contribution to the Hsp70 heat shock response at multiple stages, this factor has not acquired the regulatory support of an Hsf1, which has been demonstrated to have more than 10 times larger effects than those reported here on transcription (and protein production) of major heat shock genes.

2) The incomplete binding curve of Figure 2, which does not show saturation, is only able to provide an upper estimate of eEF1A1 affinity for Hsp70 promoter of >300nM. So eEF1A1 is a very weak binder relative to a typical sequence specific DNA binding factor, which would be useful to emphasize for the reader. I agree that there seems to be some specificity from various controls, and the new Figure 4—figure supplement 1 helps.

3) While the data support the conclusion that KD of eEF1A1 causes a ∼2-fold decrease in Hsp70 mRNA and a striking decrease in nuclear export, the case made for eEF1A1 having a role on loading Hsp70 mRNA on polysomes is considerably weaker (Figure 5). The effect on Hsp70 mRNA in polysomes could be a consequence of lower cytoplasmic Hsp70 mRNA, and the difference between light and heavy is modest especially when considering error bars. Therefore, the authors should drop or tone down the conclusion that eEF1A1 affects Hsp70 translation.

4) It's interesting that, with eEF1A1 affecting significantly so many processes in Hsp70 expression, the overall reduction in Hsp70 protein is only about 70% of the si:NT control (Figure 1—figure supplement 2). The authors should discuss this quantitatively and evaluate what this may mean about mechanisms governing this multistep pathway.

---

## [Author Response]

[Editors’ note: the author responses to the first round of peer review follow.]

Reviewer #1:

*[…] The following issues should be addressed prior to publication*.

*1) In RNAi experiments such as those performed in this manuscript, it is essential to show that similar results are obtained using two unrelated siRNAs (or shRNAs) against the same target gene to rule out off-target effects. Ideally, this should be done for all experiments in the manuscript, but minimally this needs to be done for the key experiments in the manuscript from which the main conclusions are drawn*.

We added new data to the results of most of the reported experiments in which we used alternative siRNAs to target eEF1A1:

A) The previous version of the manuscript included two other unrelated siRNAs to target mouse eEF1A1 in the western-blot quantifications (Figure 1—figure supplement 3 in the current version). These experiments were performed in parallel with the inhibition of eEF1A2, which were also performed with two unrelated siRNA (Figure 1—figure supplement 4 in the current version). The sequences of the siRNAs are indicated in the figure legend.

B) To exclude the possibility of an off-target effect we knocked down eEF1A1 with an alternative pair of siRNAs (referred to in the manuscript as Pair B). We observed similar effects in the transcription of the HSP70 gene, the stability of the Hsp70 mRNA and the transport and levels of HSP70 protein as we reported with the original pair of siRNAs (Pair A). We added these results to Figure 1—figure supplement 2, Figure 1—figure supplement 5, Figure 4—figure supplement 1, and Figure 5–figure supplement 5B. We also cite the western-blots in Figure 1—figure supplement 2 in response to comment 2 from reviewer 2. These additional data support the main conclusions of the manuscript and exclude the possibility of an off-target effect.

The sequences of the siRNAs are indicated in the materials section as Pair A and Pair B.

*2)*
Figure 1*. It would be nice to see the results minus and plus heat shock*.

We added the results of the non-heat shock conditions to Figure 1.

*3)*
Figure 2*. Have the authors performed these in vitro DNA binding experiments with purified HSF and eEF1A (*Figure 2
*uses only purified eEF1A)*.

Yes. In the paper where we first described the eEF1A/HSF1 complex we reported this experiment (Shamovsky et al., Nature (2006); Figure 2). The conclusion of that experiment was that eEF1A enhanced the *in vitro* binding of HSF1 to the heat shock element.

*4)*
Figure 2*. Isn't the fold-increase following heat-shock expected to be larger? Is there a comparable experiment for HSF1?*

The comparable experiment for HSF1 is shown in Figure 2. The fold increase of HSF1 is greater than that of eEF1A1. One possible explanation is that the occupancy of HSF1 on the HSP70 locus is restricted to the promoter at the HSE element. In the case of eEF1A, the fold-increase is observed throughout the HSP70 locus rather than at a restricted, discrete region (Figure 3 and Figure 3—figure supplement 4). The restricted localization of HSF1 on HSE results in a higher fold increase in the ChIP experiments, whereas the more mobile eEF1A1 shows a lesser increase after heat shock.

*5)*
Figure 3*. Why isn't time zero shown*?

We apologize for inadvertently omitting labeling the Y-axes in Figure 3. It has been added. In this plot we show the % of HSP70 TSs with a positive signal for eEF1A1 at HS times at which we detect active HSP70 transcription. Transcription of HSP70 at time zero (control growth conditions) was shown in Figure 1—figure supplement 6. As expected, HSP70 TSs was detected only in a small fraction of cells under non-HS conditions. We did not detect a positive signal for eEF1A in these few cells.

*6)*
Figure 3. *Isn't this experiment better suited to*
Figure 1*. Why does DRB cause decreased occupancy of RNA pol II and eEF1A at the promoter?*

Figure 3 shows the values for RNAPII phosphorylated on serine 2 (RNAPIIS2). As DRB inhibits the phosphorylation of serine 2, we observed a decrease of RNAPIIS2 over the HSP70 gene. At the promoter region, the values for RNAPIIS2 after HS-DRB treatment are similar to those of non-HS conditions. The values for total RNAPII were shown in Figure 3—figure supplement 4. In this case, the values at the promoter after DRB-HS treatment are higher than those under non-HS conditions, indicating an accumulation of RNAPII at the HSP70 promoter. Most likely this result is due to the inability of RNAPII to elongate after DRB treatment. In the case of eEF1A, we observed a decreased occupancy within the HSP70 gene. As we indicated in the text, this result together with the fact that eEF1A1 co-IPs with hypophosphorylated RNAPII (Figure 3), argues that eEF1A1 interacts with the elongation complex during HSP70 transcription.

*7)*
Figure 4*. What does relative on the Y-axis mean*?

It means relative to total HSP70 mRNA, which we have now indicated.

*8)*
Figure 4*. Has HSF been tested in the RNA-IP assay*.

HSF1 has been tested in the RNA-IP assay. A very low enrichment was observed for the HSP70 mRNA fraction bound to HSF1, which were most probably non-specific and the result of the treatment of the samples during the RNA-IP.

*9)*
Figure 4*. It would be nice to see the eEF1A binding site in the 3' UTR further localized*.

It is known that eEF1A1 binds to a stem loop region in West Nile Virus genomic RNA (Blackwell et al., 1997). Therefore, we used an m-fold structure of the HSP70 3’UTR to make deletion constructs of various stem loop regions. We deleted parts of the three stem loops one by one. The deletion of SL2 abolished binding of eEF1A1 to the 3’UTRΔSL1ΔSL2 and we were able to bind eEF1A1. This new result is added to Figure 4—figure supplement 1.

*10)*
Figure 4*. It is not clear why the promoter is needed here. Doesn't the in vitro RNA binding data of*
Figure 4
*show that eEF1A can recognize and bind directly to the 3' UTR*?

We used the different promoters to demonstrate that eEF1A1 recruited to the HSP70 promoter migrates to and binds directly to the 3’UTR (Figure 4). This binding is important for stabilizing HSP70 mRNA after heat shock. Using the foreign SV40 promoter eliminates both the promoter-specific recruitment of eEF1A1 and also the effect of eEF1A1 on luciferase expression, i.e. HSP70 mRNA stability.

*11) For several of the panels (*Figures 3, 4 and 5*) statistics need to be provided*.

For Figure 3 the p values were presented in the text: “Interestingly, the eEF1A1 occupancy inside HSP70 locus was reduced (p=0.0585) as it was for total RNAPII (p=0.0316)”.

For Figures 4 and 5 the values are not significant.

Reviewer #2:

*The manuscript by Vera et al claims the translation factor eEF1A participates at multiple steps in the human and mouse heat shock response (HSR): recruiting the HSF1 regulator to the promoter, associating with elongating Pol II and the 3'UTR of HSp70 mRNA, stabilizing the mRNA, and facilitating its export from the nucleus and to active ribosomes. These extraordinary claims, which in effect state that, “eEF1A orchestrates the entire heat shock response”, require extraordinary evidence. The authors provide a lot of experimental results in their effort to support all of these eEF1A1 functions. Some experiments are well-controlled and show reasonably convincing effects. Others are much less convincing*.

In the current manuscript we focus on HSP70 (Title: eEF1A1 couples HSP70 transcription to translation during the heat shock response); we do not state anywhere that “eEF1A1 orchestrates the entire heat shock response” as quoted by the reviewer. In the Conclusion we state that eEF1A1 synchronizes the expression of HSP70 mRNA from transcription to translation, for which we believe we provide sufficient evidence in this manuscript.

We modified one paragraph of the discussion to avoid a potential overstatement:

“The influence of eEF1A1 on several heat shock inducible genes, including HSP27 supports the notion that it controls the expression of HSF1-dependent genes in general”.

It now reads: “It is likely that eEF1A1 modulates the expression of many genes controlled by HSF1. Our results with HSP27 and other HSPs support this hypothesis (Figure 1—figure supplement 2, Figure 1–table supplement 1)”.

*How can a single 50 kDa factor do all of this*?

We believe that the importance of this manuscript is precisely because it demonstrates that one protein can perform several functions in parallel, thereby explaining the synchrony and robustness of the HSP70 expression under heat shock in mammals. As we indicate in the Discussion (end of paragraph 1), only a few factors have been shown to determine the ultimate fate of mRNAs. They have been described only in yeast, are not related to HSR, and are also known to perform other cellular functions.

*I am concerned that other general mechanisms, which do not require a direct mechanistic role in each of these very specific steps, might account for the numerous effects seen. Could the depletion of such a super abundant protein (1-3% of total cytosolic proteins) lead to general pleiotropic effects that make general cellular mechanisms sluggish*?

eEF1A1 is indeed highly abundant. However, the nuclear fraction of eEF1A1, which is responsible for the effects we reported in the manuscript, is small. We show that depletion of eEF1A1 only affects heat shock genes (transcription activation, mRNA stability, and transport), and not other unrelated genes or processes. We also show that depletion of the second isoform of eEF1A (which is 95% identical to eEF1A1 and can clearly fulfill, by itself, the translational requirements of a cell as it is the only eEF1A isoform expressed in muscle cells and neurons) does not have any effect on HSR. This second isoform is highly expressed in various cancer cell lines, including HeLa and NSC34 - cells, which we used in the current study (Figure 1—figure supplement 3) to demonstrate that knock down of EF1A2, which has the same general properties as eEF1A1 with respect to translation, both *in vitro* and *in vivo*, does not alter the HSR.

*eEF1A1 is also known to have chaperone function (see J. Biol. Chem. 2007, 282:4076), and its depletion might have a broad array of effects including those seen here. The previously documented role of eEF1A1 by this group as having a role in trimerization and activation of HSF1 could be consistent with such a chaperone role*.

The chaperone function of eEF1A1 homologs has been reported in bacteria, yeast and for the mammalian mitochondrial translation elongation factor Tu (see J. Biol. Chem. 2007, 282:4076). We don’t believe this is damaging to our conclusions, and, indeed, we agree that eEF1A does function as a chaperone in promoting HSF1 trimerization, as proposed in our earlier report (42). However, the additional functions of eEF1A1 that we currently describe with respect to HSR cannot be explained by the same chaperone activity.

*These general concerns and the lack of a mapping of specific functions to distinct parts or mutations of the protein, lead me to the view that this grand view of eEF1A is too ambitious for a single paper and for the data in hand. This is already a bulky paper and it may be better suited being published as a couple of more focused papers in more specialized journals than eLife. However, publication either in eLife or elsewhere requires addressing these general concerns and specific concerns below*.

Mapping specific functions to distinct regions of eEF1A is the next step in understanding the molecular mechanism of eEF1A1-mediated HSR. This work is currently underway and beyond the scope of the current manuscript, which is already “bulky”, as noted by the reviewer, and which elucidates how HSP70 expression is synchronized during heat shock by the multiple functions of eEF1A1. This primary work needs to be published before we subsequently address the biochemical details of eEF1A1 function.

We hope that the above clarifications address the general concerns of this reviewer.

*Specific concerns*:

*1) The claim that translation is not affected by knockdown of eEF1A1 is not adequately supported by the data. The authors show that total translation is unaffected in cells where eEF1A1 was reduced to ∼35% of normal levels (*Figure 1—figure supplement 1*). The heat shock for this experiment was done at 40C instead of 43C. All the other experiments were done at full heat shock temperatures 43C or 45C. The 40C appears not to be a typo, because, under their 40C heat shock, there is not the change in protein patterns expected for a normal heat shock. A broad effect on general translation in full heat shock conditions caused by.eEF1A1 could lead to reduced HSPs*.

General translation is shutdown under full heat shock conditions, so the effect of eEF1A on general protein synthesis cannot be determined. Accordingly, we used 40^°^C to show that a mild heat-shock equally decreases general translation of mock-transfected cells and si:eEF1A1 transfected cells.

*2) With no standard curve to show their Western assays are linear, it is inappropriate to ascribe the accuracy implied by a statement “knock down of eEF1A1 <38%” and in other numbers in*
Figure 1
*and supplemental figures. It's not sufficient to normalize to a GAPDH band*.

In Figure 1 we referred to a representative western-blot and its quantifications. The reviewer is correct that the accuracy implied in the quoted statement cannot rely solely on those numbers. We therefore performed further quantification.

All the western-blots presented were performed using chemiluminescence and X-ray films. Regardless of how accurate the film quantifications are, we always observe an obvious decrease in the levels of HSP70 protein in different cell lines from different species using different siRNAs to target eEF1A1 (see Figure 1, Figure 1—figure supplement 2, for representative blots prepared from different human cell lines:, quantification of 3 independent western blots with two unrelated si:eEF1A1).

To address the reviewer’s concern in full, we added two new western blots using an alternative set of siRNAs against eEF1A1 (Pair B) to exclude potential off-target effect (Figure 1—figure supplement 2). The quantitation was performed as follows:

A) Li-COR Odyssey Imager for direct detection. With this fluorescent system western blot detection of HSP70 is reported to be quantitative across 4.3 orders of magnitude, from 5 pg to 100 ng (http://www.licor.com/bio/applications/quantitative_western_blots/quantification.html).

B) We used two induction times for HSP70 protein expression (2 and 6 hours of recovery after 1 hour of heat shock) not to saturate the system.

C) We included ponceau S staining of the membrane to demonstrate that our normalizations to GAPDH accurately represents the total amount of proteins loaded in each condition.

D) We quantified the two western blots, and in both cases we observed ∼70% reduction of HSP70 protein levels upon heat shock in cells knocked down of eEF1A1. In these cells levels of eEF1A were reduced to 25%. Concurrently, eEF1A1 mRNA levels were reduced to 20%, as quantified by RT-QPCR.

The numerical values obtained by Li-COR are comparable to the values we previously presented. Therefore, we have substituted the numbers indicated in the main text by saying “to a third of the normal levels” or to ∼70 % as previously indicated.

*3)*
Figure 1
*needs more controls*. *Is the complete loss of the red eEF1A only in the cytoplasm due to a 65% knock down of this protein??*

The cells represented Figure 1 stably express Cherry-Flag-eEF1A1. The loss of the cherry signal in the cytoplasm was indeed due to the knock down of the eEF1A1. The loss of eEF1A1 occurs not only in the cytoplasm; it is also proportionately reduced in the nucleus. The exposure time determines the intensity of signal. The remaining cherry fluorescence, due to the remaining 35% eEF1A1, is less visible due to optimization of the exposure to capture the full expression of eEF1A1. The remaining Cherry-Flag-eEF1A1 fluorescence is more easily observed on a computer screen.Author response image 1.Immofluorescence of cells expressing Cherry-flag-eEF1A1 growing at 37^°^C. Mock transfected cells (Si:NT and cells knocked down for eEF1A1 (si:eEF1A1) were immunostained with an antibody specific for eEF1A1 (Millipore, # 05-235) and a secondary antibody (Invitrogen, goat anti mouse (alexa 647)) (red). Nucleus was stained with dapi (blue).

*4) Are there no error bars associated with the control in*
Figure 1—figure supplement 2 and Figure 1—figure supplement 3*? If there are bars in proportion to the measurement of eEF1A1, then are the differences as significant as the authors’ state*?

To compare three biological replicates for each experiments all values were normalized to si:NT. NT values were defined as 1 and, therefore, have no error bars, but values for si:eEF1A samples do have error bars. The differences in values are statistically significant, as indicated.

*5) With eEF1A1 being so abundant, can many of the interactions observed with other proteins and RNAs in the cell be non-specific or low affinity (*Figure 2*). Perhaps more quantification and use of other proteins as controls that are similarly abundant would help*.

As explained below, we believe there are enough controls to demonstrate that interactions of eEF1A1 with mRNA, DNA, and proteins are specific:

For the interaction of eEF1A1 with mRNA: if the reported interactions depended on the relative abundance we should have detected a higher interaction of eEF1A1 with GAPDH mRNA than with HSP70 mRNA, because GAPDH mRNA is substantially more abundant. In our RNA IP experiment (Figure 4) we detected only an eEF1A-HSP70 mRNA interaction. We included PABP, which is less abundant but interacts with both mRNAs, as a control in this same experiment. Finally, we have even identified a specific region within the 3’UTR of HSP70 mRNA that interacts with eEF1A1 (see new Figure 4—figure supplement 1)

For the interaction of eEF1A1 with DNA: we used GAPDH as a negative control. If the detection of eEF1A on the DNA were non-specific we should have observed the same enrichment for the GAPDH promoter and ORF as for the HSP70 gene. We also used Flag to ChIP the exogenous protein, which is less abundant than its endogenous counterpart (Figure 1—figure supplement 6) and we obtained similar results as with the endogenous protein (Figure 3 and Figure 3—figure supplement 4; cells lacking Cherry-Flag-eEF1A1 were used as a negative control.

For the interaction of eEF1A1 with other proteins: we provide data on interactions only with HSF1 and PolII. Endogenous controls are provided for both of the IPs. We detect eEF1A1-HSF1 and eEF1A1-PolII interactions only under heat shock conditions (Figure 2 and Figure 3).

If all these interactions were non-specific due the high cellular abundance of eEF1A1, we should also have detected them under non-HS conditions, which did not occur. In other words, the specificity of each interaction under HS conditions is supported by the corresponding non-interaction control under non-heat shock conditions.

*6) In the binding of eEF1A1 to Hsp70 RNA, a single high concentration of protein is used. A binding curve and estimate of the Kd should be provided, if this is claimed to be an RNA binder*.

This result is now shown in Figure 4—figure supplement 1.

*7) It was recently reported that inhibition of translation inactivates Hsf1 (Santagata et al. Science, 2013). If the translation is found to be compromised in eEF1A1 knockdown cells, then many observations reported in this manuscript can be explained by the possibility of unaccounted effect of eEF1A1 knockdown in inactivation of Hsf1 and hence, attenuation of Hsp70 and other heat shock proteins expression upon heat shock*.

We do not find that translation is compromised in eEF1A1 knockdown cells (Figure 1—figure supplement 1). Although this observation alone should suffice in resolving the reviewer’s concern, we address it below in more detail:

Santagata et al. show that there is a tight coordination of protein translation and HSF1 activation that specifically supports the anabolic malignant state. To reach this conclusion they used a translation initiation inhibitor RHT. This drug has a strong effect in a panel of human cancer cell lines but much less of an effect in non-tumorigenic cells. In our experiments we used human and mouse fibroblast that are non-tumorigenic (TIGs and MEFs). We used the MDA-MB-231 breast cancer cell line (not used by Santagata et al.) and obtained similar results as with non-tumorigenic cell lines. We used the MDA-MB-231 cell line because under non-heat shock condition these cells show very little expression of HSP70 mRNA.

The experiments by Santagata et al. were performed under non-heat shock conditions, whereas most of our experiments were performed upon heat-shock. Without heat shock their tumorigenic cells continued to display a high level of HSP70 mRNA and efficient binding of HSF1 to the HSP70 promoter. This binding disappears when they add RHT. In contrast, we did not detect any strong HSF1 binding to the HSP70 promoter under control (non-HS) conditions. Moreover, this weak binding was not altered when we knocked down eEF1A1 (New Figure 2). Interestingly, it is well established and demonstrated in the field that both translation initiation and elongation are already shut down during heat shock conditions when HSF1 shows its higher activity.

The only cell line the two groups used in common is HeLa. We used HeLa because they express high levels of both eEF1A isoforms, eEF1A1 and eEF1A2. Although both isoforms share the same role in translation elongation, only when we knocked down eEF1A1 did we observe a decrease in HSP70 mRNA (Figure 1—figure supplement 3). Therefore, the role of eEF1A1 during the heat shock response is independent of its role in translation.

For all the reasons indicated above we believe that there is not an unaccounted effect of eEF1A1 knockdown in the inactivation of HSF1 under the conditions we used for the experiments presented in our current manuscript.

*8) Finally, is the proposed role of eEF1A1 in recruitment and activation of Hsf1 limited to major Heat Shock Protein genes or is it much broader and applicable for all Hsf1 regulated genes? Evidence exists for Hsf1 regulating expression of genes other than HSPs (Trinklein et al., 2004 & Mendilli et al., 2012) and I think it would be quite informative, if authors could include few of these genes and examine the role of eEF1A1 knockdown in their induction upon heat shock*.

This is an important question, which we will address in a future report. In the current manuscript we focus primarily on HSP70 in order to characterize each step in its induction pathway: transcription, mRNA stability, transport and translation. To determine whether every gene regulated by HSF1 is also subject to regulation by eEF1A1 is beyond the limits of any single manuscript. We have added an mRNA profiler in Figure 1–table supplement 1 and discussed the likely possibility of the role of eEF1A1 in regulating the expression of heat shock genes other than HSP70. Finally, to address the reviewer’s question in full, different cell lines (from non-tumorigenic to highly malignant) would need to be analyzed, because HSF1 drives a transcriptional program distinct from heat shock to specifically support highly malignant human cancers (Mendillo et al. 2012).

[Editors’ note: the author responses to the re-review follow.]

*1) The previous Impact Statement stated “eEF1A1 orchestrates the entire process of the heat shock response, from transcription activation of HSP70 gene, to HSP70 mRNA stabilization, nuclear export, and translation”. I apologize for abbreviating this to “eEF1A orchestrates the entire heat shock response”. The current Impact statement now is “eEF1A1 orchestrates the process of heat shock response, from transcription activation of the HSP70 gene, to HSP70 mRNA stabilization, nuclear export, and translation”. The word “entire” is now deleted, which helps, but this is still an overstatement, especially considering the complexity of the heat shock response. First, the “process of the heat shock response” is too broad, as this study focuses on Hsp70. Second, the work “orchestrates” gives eEF1A1 a status of a major regulator like HSF1. While the paper provides data supporting an eEF1A1 contribution to the Hsp70 heat shock response at multiple stages, this factor has not acquired the regulatory support of an Hsf1, which has been demonstrated to have more than 10 times larger effects than those reported here on transcription (and protein production) of major heat shock genes*.

As requested, we changed the word “orchestrate” in the Impact Statement to less prominent “modulate”. It now reads: *“eEF1A1* modulates the process of heat shock response, from transcription activation of the HSP70 gene, to HSP70 mRNA stabilization, nuclear export, and translation”.

*2) The incomplete binding curve of*
Figure 2*, which does not show saturation, is only able to provide an upper estimate of eEF1A1 affinity for Hsp70 promoter of >300nM. So eEF1A1 is a very weak binder relative to a typical sequence specific DNA binding factor, which would be useful to emphasize for the reader. I agree that there seems to be some specificity from various controls, and the new*
Figure 4—figure supplement 1
*helps*.

As suggested we now emphasize a relatively weak DNA binding capability of eEF1A1: “Moreover, in vitro binding of recombinant eEF1A1 to a DNA fragment of HSP70 promoter suggests direct and specific, albeit a relatively weak, interaction of eEF1A1 to HSP70 promoter (Figure 2). We further confirmed the presence of eEF1A1 at HSP70 locus by ChIP-QPCR… experiments ”

*3) While the data support the conclusion that KD of eEF1A1 causes a ∼2-fold decrease in Hsp70 mRNA and a striking decrease in nuclear export, the case made for eEF1A1 having a role on loading Hsp70 mRNA on polysomes is considerably weaker (*Figure 5*). The effect on Hsp70 mRNA in polysomes could be a consequence of lower cytoplasmic Hsp70 mRNA, and the difference between light and heavy is modest especially when considering error bars. Therefore, the authors should drop or tone down the conclusion that eEF1A1 affects Hsp70 translation*.

As suggested, we toned down the conclusion regarding eEF1A1 affecting HSP70 translation by doing the following changes in the text:

Abstract: “…facilitating its export from the nucleus to the active ribosomes” to “facilitating its nuclear export”.

also: “…eEF1A1 facilitates HSP70 mRNA nuclear export and translation” to “..eEF1A1 facilitates HSP70 mRNA nuclear export”

and: “… indicating that eEF1A1 helped the loading of HSP70 into polysomes” to “… suggesting that eEF1A1 helped the loading of HSP70 mRNA into polysomes.”

*4) It's interesting that, with eEF1A1 affecting significantly so many processes in Hsp70 expression, the overall reduction in Hsp70 protein is only about 70% of the si:NT control (*Figure 1—figure supplement 2*). The authors should discuss this quantitatively and evaluate what this may mean about mechanisms governing this multistep pathway*.

As suggested, we now expand the Discussion to address this comment:

“…We observed that each of these steps gets impaired when eEF1A1 is partially depleted. It is the sum of the effects on HSP70 mRNA synthesis, stability and transport that accounts for a ∼70% reduction of the HSP70 protein level and decreased thermotolerance in cells partially knocked down for eEF1A1. It is likely that the remaining eEF1A1 supports the residual HSP70 expression.”

Clarification for the editor/reviewer: We would like to emphasize that 70% reduction of HSP70 is not a low value as implied by the reviewer’s term “only”: The western blot experiments were done from a pool of cells treated with eEF1A1 siRNA. The efficiency of transfection by siRNAs and eEF1A1 inhibition was less than 75%. Thus, in any given experiment there are some cells responding to heat shock normally by producing HSP70 protein. This should account at least for some of the remaining HSP70 produced under the eEF1A1 depletion conditions. Moreover, a 70% reduction in HSP70 protein levels is biologically significant since there is a decrease in cell thermotolerance comparable with that of HSF1 depletion (Figure 1).